# A general model-based causal inference method overcomes the curse of synchrony and indirect effect

Se Ho Park[1,2], Seokmin Ha[2,3] & Jae Kyoung Kim [ORCID][2,3] ✉

To identify causation, model-free inference methods, such as Granger Causality, have been widely used due to their flexibility. However, they have difficulty distinguishing synchrony and indirect effects from direct causation, leading to false predictions. To overcome this, model-based inference methods that test the reproducibility of data with a specific mechanistic model to infer causality were developed. However, they can only be applied to systems described by a specific model, greatly limiting their applicability. Here, we address this limitation by deriving an easily testable condition for a general monotonic ODE model to reproduce time-series data. We built a user-friendly computational package, General ODE-Based Inference (GOBI), which is applicable to nearly any monotonic system with positive and negative regulations described by ODE. GOBI successfully inferred positive and negative regulations in various networks at both the molecular and population levels, unlike existing model-free methods. Thus, this accurate and broadly applicable inference method is a powerful tool for understanding complex dynamical systems.

Identifying causal interaction is crucial to understand the underlying mechanism of systems in nature. A recent surge in time-series data collection with advanced technology offers opportunities to computationally uncover causation[1]. Various model-free methods, such as Granger causality (GC)[2] and convergent cross mapping (CCM)[3], have been widely used to infer causation from time-series data. Although they are easy to implement and broadly applicable[4–10], they usually struggle to differentiate generalized synchrony (i.e., similar periods among components) versus causality[11–15] and distinguish between direct and indirect causation[16–20]. For instance, when oscillatory time-series data is given, nearly all-to-all connected networks are inferred[12]. To prevent such false positive predictions, model-free methods have been improved (e.g., partial cross mapping (PCM)[20]), but further investigation is needed to show their universal validity.

Alternatively, model-based methods infer causality by testing the reproducibility of time-series data with mechanistic models using various methods such as simulated annealing[21] and the Kalman Filter[22,23]. Although testing the reproducibility is computationally expensive, as long as the underlying model is accurate, the model-

based inference method is accurate even in the presence of generalized synchrony in time series and indirect effect[21–29]. However, the inference results strongly depend on the choice of model, and inaccurate model imposition can result in false positive predictions, limiting their applicability. To overcome this limit, inference methods using flexible models were developed[30–39]. In particular, the most recent method, ION[12], infers causation from $X$ to $Y$ described by the general monotonic ODE model between two components, i.e., $\frac{dY}{dt} = f(X,Y)$. However, ION is applicable only when every component is affected by at most one another component.

Here, we develop a model-based method that infers interactions among multiple components described by the general monotonic ODE model:

$$\frac{dY}{dt} = f(\mathbf{X}) = f(X_1, X_2, \cdots, X_N), \tag{1}$$

where $f$ can be any smooth and monotonic increasing or decreasing functions of $X_i$ and $X_N$ is $Y$ in the presence of self-regulation. Thus, our

[1]Department of Mathematics, University of Wisconsin-Madison, Madison, WI 53706, USA. [2]Biomedical Mathematics Group, Institute for Basic Science, Daejeon 34126, Republic of Korea. [3]Department of Mathematical Sciences, KAIST, Daejeon 34141, Republic of Korea. ✉e-mail: jaekkim@kaist.ac.kr

approach considerably resolves the fundamental limit of model-based inference: strong dependence on a chosen model. Furthermore, we derive a simple condition for the reproducibility of time series with Eq. (1), which does not require computationally expensive fitting, unlike previous model-based approaches. To facilitate our approach, we develop a user-friendly computational package, GOBI (General ODE-Based Inference). GOBI successfully infers causal relationships in gene regulatory networks, ecological systems, and cardiovascular disease caused by air pollution from synchronous time-series data, with which popular model-free methods fail at inference. Furthermore, GOBI can also distinguish between direct and indirect causation, even from noisy time-series data. Because GOBI is both accurate and broadly applicable, which had not been achieved by previous model-free or model-based inference methods, it can be a powerful tool in understanding complex dynamical systems.

## Results

### Inferring regulation types from time series

We first illustrate the common properties of time series generated by either positive or negative regulation with simple examples. When the input signal $X$ positively regulates $Y$ ($X \rightarrow Y$) (Fig. 1a), $\dot{Y}$ increases whenever $X$ increases. Thus, for any pair of time points $t$ and $t^*$ with which $X^d(t, t^*) := X(t) - X(t^*) > 0$, $\dot{Y}^d(t, t^*) := \dot{Y}(t) - \dot{Y}(t^*) > 0$. Similarly, when $X$ negatively regulates $Y$ ($X \dashv Y$) (Fig. 1c left), if $X^d(t, t^*) < 0$, then $\dot{Y}^d(t, t^*) > 0$. Thus, in the presence of either positive ($\sigma = +$) or negative ($\sigma = -$) regulation, the following *regulation-detection function* is always positive (Fig. 1b, c):

$$I_{X^\sigma}^Y(t, t^*) := \sigma X^d(t, t^*) \cdot \dot{Y}^d(t, t^*) \tag{2}$$

defined on $(t, t^*)$ such that $\sigma X^d(t, t^*) > 0$.

This idea can be extended to a case with multiple causes. For instance, when $X_1$ and $X_2$ positively regulate $Y$ together (Fig. 1d), if both $X_1^d > 0$ and $X_2^d > 0$, then $\dot{Y}^d > 0$. This leads to the positivity of the regulation-detection function for $\begin{matrix} X_1 \rightarrow \\ X_2 \rightarrow \end{matrix} Y$, $I_{X_1 X_2}^Y(t, t^*) := X_1^d(t, t^*) \cdot X_2^d(t, t^*) \cdot \dot{Y}^d(t, t^*)$, defined for $(t, t^*)$ such that $X_1^d(t, t^*) > 0$ and $X_2^d(t, t^*) > 0$ (Fig. 1e). Similarly, if $X_1$ and $X_2$ positively and negatively regulate $Y$, respectively ($\begin{matrix} X_1 \rightarrow \\ X_2 \dashv \end{matrix} Y$) (Fig. 1g), the regulation-detection function for

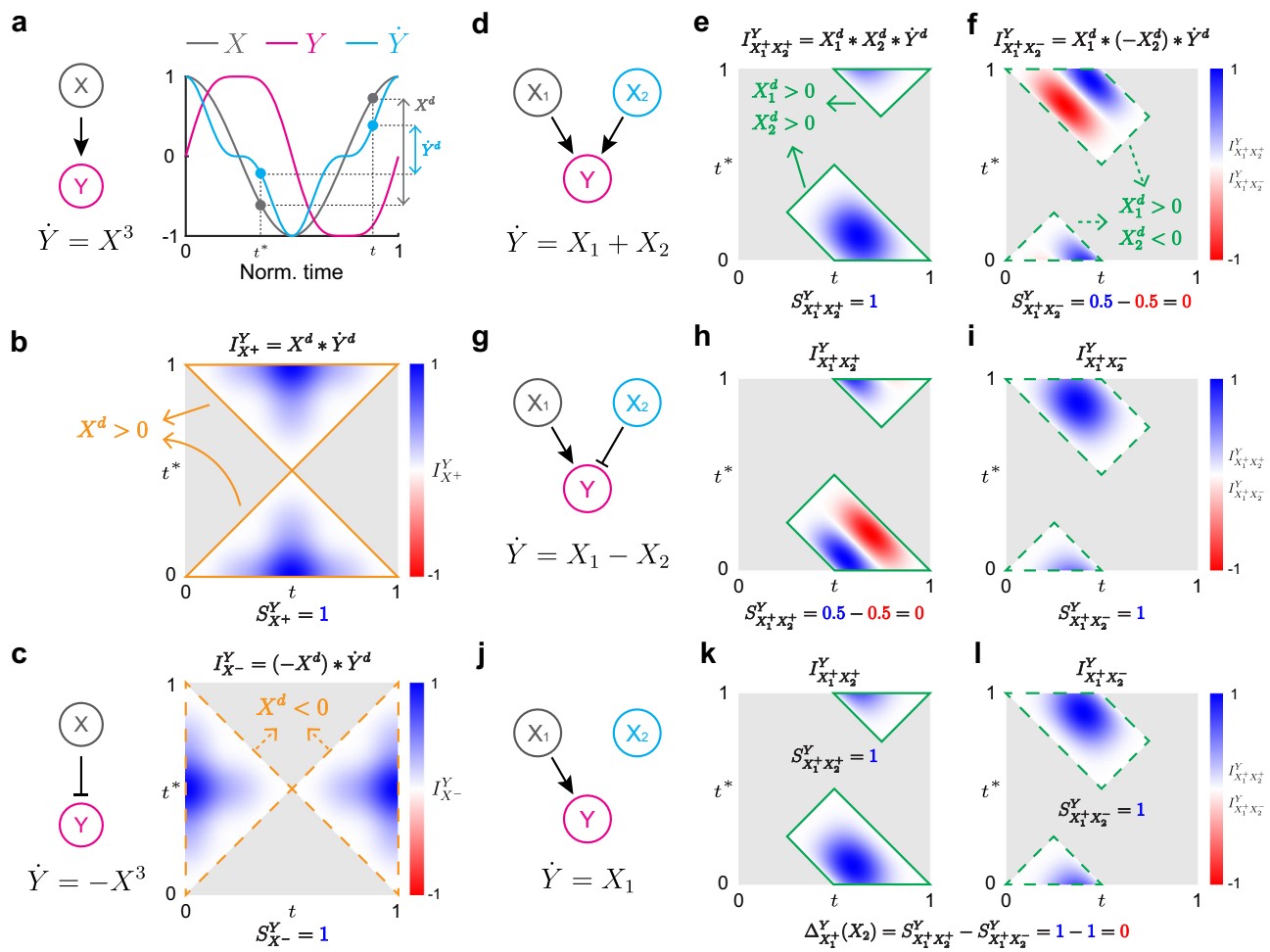

**Fig. 1 | Inferring regulation types using regulation-detection functions and scores. a** Because $X$ positively regulates $Y$, as $X$ increases, $\dot{Y}$ increases. Thus, whenever $X^d(t, t^*) = X(t) - X(t^*) > 0$, $\dot{Y}^d(t, t^*) = \dot{Y}(t) - \dot{Y}(t^*) > 0$. **b** Therefore, when $X^d(t, t^*) > 0$, the regulation-detection function $I_{X^+}^Y(t, t^*) := X^d(t, t^*) \cdot \dot{Y}^d(t, t^*)$ is always positive. Here, $I$ is in the range $[-1, 1]$ since all the time series are normalized. **c** If $X$ negatively regulates $Y$, $I_{X^-}^Y := (-X^d) \cdot \dot{Y}^d$ is always positive when $X^d(t, t^*) < 0$. **d–i** When $X_1$ and $X_2$ positively regulate $Y$, as $X_1$ and $X_2$ increase ($X_1^d > 0$, $X_2^d > 0$), $\dot{Y}$ increases ($\dot{Y}^d > 0$) (**d**). Thus, when $X_1^d(t, t^*) > 0$ and $X_2^d(t, t^*) > 0$, $I_{X_1^+ X_2^+}^Y := X_1^d \cdot X_2^d \cdot \dot{Y}^d$ is positive

(**e**). When $X_1$ and $X_2$ positively and negatively regulate $Y$, respectively (**g**), $I_{X_1^+ X_2^-}^Y := X_1^d \cdot (-X_2^d) \cdot \dot{Y}^d$ is always positive when $X_1^d(t, t^*) > 0$ and $X_2^d(t, t^*) < 0$ (**i**). Such positivity disappears for the regulation-detection functions, which do not match with the actual regulation type (**f, h**). **j–l** When $X_1$ positively regulates $Y$ and $X_2$ does not regulate $Y$ (**j**), both $I_{X_1^+ X_2^+}^Y := X_1^d \cdot X_2^d \cdot \dot{Y}^d$ (**k**) and $I_{X_1^+ X_2^-}^Y := X_1^d \cdot (-X_2^d) \cdot \dot{Y}^d$ (**l**) are positive because the regulation type of $X_2$ does not matter. Here, we use $X_1(t) = \cos(2\pi t)$ and $X_2(t) = \sin(2\pi t)$ as the input signal and $Y(0) = 0$ for simulation on $[0, 1]$. Source data are provided as a Source Data file.

$\begin{matrix} X_1 \to \\ X_2 \dashv \end{matrix} Y$, $I^Y_{X_1^+ X_2^-}(t,t^*) := X_1^d(t,t^*) \cdot (-X_2^d(t,t^*)) \cdot \dot{Y}^d(t,t^*)$, is positive for $(t,t^*)$ such that $X_1^d(t,t^*) > 0$ and $X_2^d(t,t^*) < 0$ (Fig. 1i). Note that unlike $I^Y_{X_1^+ X_2^+}$ ($I^Y_{X_1^+ X_2^-}$), $I^Y_{X_1^+ X_2^-}$ ($I^Y_{X_1^+ X_2^+}$) is not always positive for $\begin{matrix} X_1 \to \\ X_2 \to \end{matrix} Y$ ($\begin{matrix} X_1 \to \\ X_2 \dashv \end{matrix} Y$) (Fig. 1f, h). See Supplementary Fig. 1 for other types of 2D regulations.

In the presence of monotonic regulation, the regulation-detection function $I^Y_{\mathbf{X}^\sigma}$ is positive. The positivity of the $I^Y_{\mathbf{X}^\sigma}$ can be quantified with its normalized integral, *regulation-detection score* $S^Y_{\mathbf{X}^\sigma}$ (Eq. (4)). Thus, $S^Y_{\mathbf{X}^\sigma} = 1$ in the presence of regulation type $\sigma$ since the regulation-detection function is positive (see Supplementary Information for details). However, even in the absence of regulation type $\sigma$, $S^Y_{\mathbf{X}^\sigma}$ can often be one. For instance, when $X_1$ positively regulates $Y$ and $X_2$ does not regulate $Y$ (Fig. 1j), $\dot{Y}$ increases whenever $X_1$ increases regardless of $X_2$. Thus, both $I^Y_{X_1^+ X_2^+}$ and $I^Y_{X_1^+ X_2^-}$ are positive (Fig. 1k, l). Here, $S^Y_{X_1^+ X_2^+} = S^Y_{X_1^+ X_2^-} = 1$ reflects that $X_2$ does not affect the regulation $X_1 \to Y$. Thus, to quantify the effect of a new component (e.g., $X_2$) on an existing regulation (e.g., $X_1 \to Y$), we develop a *regulation-delta function* $\Delta$:

$$\Delta^Y_{X_1^+}(X_2) := S^Y_{X_1^+ X_2^+} - S^Y_{X_1^+ X_2^-}. \tag{3}$$

If $\Delta^Y_{X_1^+}(X_2) = 0$, $S^Y_{X_1^+ X_2^+} = 1$ ($S^Y_{X_1^+ X_2^-} = 1$) does not indicate the presence of $\begin{matrix} X_1 \to \\ X_2 \to \end{matrix} Y$ ($\begin{matrix} X_1 \to \\ X_2 \dashv \end{matrix} Y$).

## Inferring regulatory network structures

$S^Y_{\mathbf{X}^\sigma} = 1$ together with $\Delta \neq 0$ can be used as an indicator of regulation type $\sigma$ from $\mathbf{X}$ to $Y$. Based on this, we construct a framework for inferring a regulatory network from time-series data (Fig. 2a). To illustrate this, we obtain multiple time-series data simulated with random input signal $A$ and different initial conditions of $B$ and $C$ randomly selected from $[-1, 1]$.

From each time series, the regulation-detection score $S^Y_{X^\sigma}$ is calculated for every type of 1D regulation $\sigma$ from $X$ to $Y$ ($X, Y = A, B,$ or $C$) (Step 1). Because only $A \dashv B$ satisfies the criteria $S^Y_{X^\sigma} = 1$ for every time series, only $A \dashv B$ is inferred as 1D regulation. Note that even for the other regulations, $S^Y_{\mathbf{X}^\sigma} = 1$ can occur for a few time series, leading to a false positive prediction. This can be prevented by using multiple time series. Next, $S^Y_{\mathbf{X}^\sigma}$ is calculated for every 2D regulation type $\sigma$ (Step 2). Three types of regulation ($\begin{matrix} A \dashv \\ C \to \end{matrix} B$, $\begin{matrix} A \dashv \\ C \dashv \end{matrix} B$, and $\begin{matrix} A \to \\ B \to \end{matrix} C$) satisfy the criteria $S^Y_{\mathbf{X}^\sigma} = 1$ for every time series. Among these, we can identify false positive regulations by using a regulation-delta function (Step 3). $\Delta^B_{A^-}(C)$ is equal to zero for every time series, indicating that $\begin{matrix} A \dashv \\ C \to \end{matrix} B$ and $\begin{matrix} A \dashv \\ C \dashv \end{matrix} B$ are false positive regulations. Thus, $\begin{matrix} A \to \\ B \to \end{matrix} C$ is the only inferred 2D regulation as it satisfies the criteria for the regulation-delta function ($\Delta^C_{A^+}(B) \neq 0$ and $\Delta^C_{B^+}(A) \neq 0$). By merging the inferred 1D and 2D regulations, the regulatory network is successfully inferred. Here, note that regulation $A \to C$ is not detected by the 1D regulation−detection score since $C$ has multiple causes. However, the 2D regulation-detection score detects $\begin{matrix} A \to \\ B \to \end{matrix} C$, which contains $A \to C$. This demonstrates the need for multi-dimensional inferences, as the 1D criteria alone would not have been sufficient to fully capture the regulatory relationships in the network. Since this system has three components, we infer up to 2D regulations. If there are $N$ components in the system, we go up to $(N-1)$D regulations (Supplementary Fig. 2).

We apply the framework to infer regulatory networks from simulated time-series data of various biological models. In these models, the degradation rates of molecules increase as their concentrations increase, like in most biological systems (i.e., self-regulation is negative).

Such prior information, including the types of self-regulation, can be incorporated into our framework. For example, to incorporate negative self-regulation, when detecting $N$D regulation, one can use the $(N+1)$D regulation-detection function and score that include negative self-regulation. Specifically, when inferring 1D positive regulation from $X$ to $Y$, the criterion $S^Y_{X^+ Y^-} = 1$ is used. To illustrate this, we assume the negative self-regulation to infer the network structures of biological models (see below for details). Note that this assumption is optional for inference (see Supplementary Information for details).

From the time series simulated with the Kim-Forger model (Fig. 2b left), describing the negative feedback loop of the mammalian circadian clock[40], using the criteria $S^Y_{X^\sigma Y^-} = 1$, two positive 1D regulations ($M \to P_C$ and $P_C \to P$) and one negative 1D regulation ($P \dashv M$) are inferred (Fig. 2b middle). Among the six different types of 2D regulations ($\begin{matrix} M \to \\ P \to \end{matrix} P_C$, $\begin{matrix} M \to \\ P \dashv \end{matrix} P_C$, $\begin{matrix} P_C \to \\ M \to \end{matrix} P$, $\begin{matrix} P_C \to \\ M \dashv \end{matrix} P$, $\begin{matrix} P \dashv \\ P_C \to \end{matrix} M$, and $\begin{matrix} P \dashv \\ P_C \dashv \end{matrix} M$) satisfying the criteria $S^Y_{\mathbf{X}^\sigma Y^-} = 1$ for all the time series, none of them pass the $\Delta$ test (i.e., $\Delta^{P_C}_{M^+}(P) = \Delta^P_{P_C^+}(M) = \Delta^M_{P^-}(P_C) = 0$) (Fig. 2b middle). Thus, no 2D regulation is inferred. By merging the three inferred 1D regulations, the negative feedback loop structure is recovered (Fig. 2b right). Our method also successfully infers the negative feedback loop structure of *Frzilator*[41] (Fig. 2c) and the 4-state Goodwin oscillator[42] (Fig. 2d). Furthermore, our framework correctly infers systems having 2D regulations: the Goldbeter model describing the *Drosophila* circadian clock[43] (Fig. 2e) and the regulatory network of the cAMP oscillator of *Dictyostelium*[44] (Fig. 2f) (see Supplementary Information for the equations and parameters of the models and Supplementary Data 1 for detailed inference results). Here, assuming negative self-regulation allows us to reduce $N$D regulation to $(N-1)$D regulation. This simplification is important for accurate inference when data is limited (Supplementary Fig. 3). Moreover, it should be noted that when the assumptions about the types of self-regulation are not met, only the links that violate these assumptions become untrustworthy, while the other inference results are not affected (Supplementary Fig. 3). Taken together, our method successfully infers regulatory networks from various in silico systems regardless of their explicit forms of ODE by assuming a general monotonic ODE (Eq. (1)). Unlike our approach, model-based methods that require specifying the model equations produce inaccurate inferences if inappropriate functional bases are chosen (Supplementary Fig. 4).

## Inference with noisy time series

In the presence of noise in the time-series data, the regulation-detection score ($S^Y_{\mathbf{X}^\sigma}$) is perturbed. Thus, $S^Y_{\mathbf{X}^\sigma}$ may not be one even if there is a regulation type $\sigma$ from $\mathbf{X}$ to $Y$. For example, in the case of an Incoherent Feed-forward Loop (IFL) which contains $A \dashv B$ (Fig. 3a), $S^B_{A^-}$ is always one in the absence of noise (Fig. 2a Step 1, blue), but not in the presence of noise (Fig. 3b blue). Thus, for noisy data, we need to relax the criteria $S^Y_{\mathbf{X}^\sigma} = 1$ to $S^Y_{\mathbf{X}^\sigma} > S^{\text{thres}}$ where $S^{\text{thres}} < 1$ is a threshold. Because $S^B_{A^-}$ gets farther away from one as the noise level increases, $S^{\text{thres}}$ should also be decreased accordingly. We choose $S^{\text{thres}}$ as $0.9 - 0.005 \times$ (noise level) with which true and false regulations can be distinguished in the majority of cases for our previous in silico examples (Fig. 3b and Supplementary Fig. 5e). For instance, $S^{\text{thres}}$ (green dashed line, Fig. 3b) overall separates true regulation (Fig. 3b blue) and false regulation (Fig. 3b red). Here we choose $A \to C$, which has the highest regulation-detection score among all false positive 1D regulations (Fig. 2a Step 1, red).

We found that the fraction of time-series data satisfying $S^Y_{\mathbf{X}^\sigma} > S^{\text{thres}}$, which we refer to as the *Total Regulation Score (TRS)* (Fig. 3c left), more clearly distinguishes the true (Fig. 3c right blue) and false (Fig. 3c right red) regulations. Thus, we use the criteria $\text{TRS}^Y_{\mathbf{X}^\sigma} > \text{TRS}^{\text{thres}}$ to infer the regulation. Similar to $S^{\text{thres}}$, $\text{TRS}^{\text{thres}}$ also decreases as the noise level increases. Specifically, we use $\text{TRS}^{\text{thres}} = 0.9 - 0.01 \times$ (noise level),

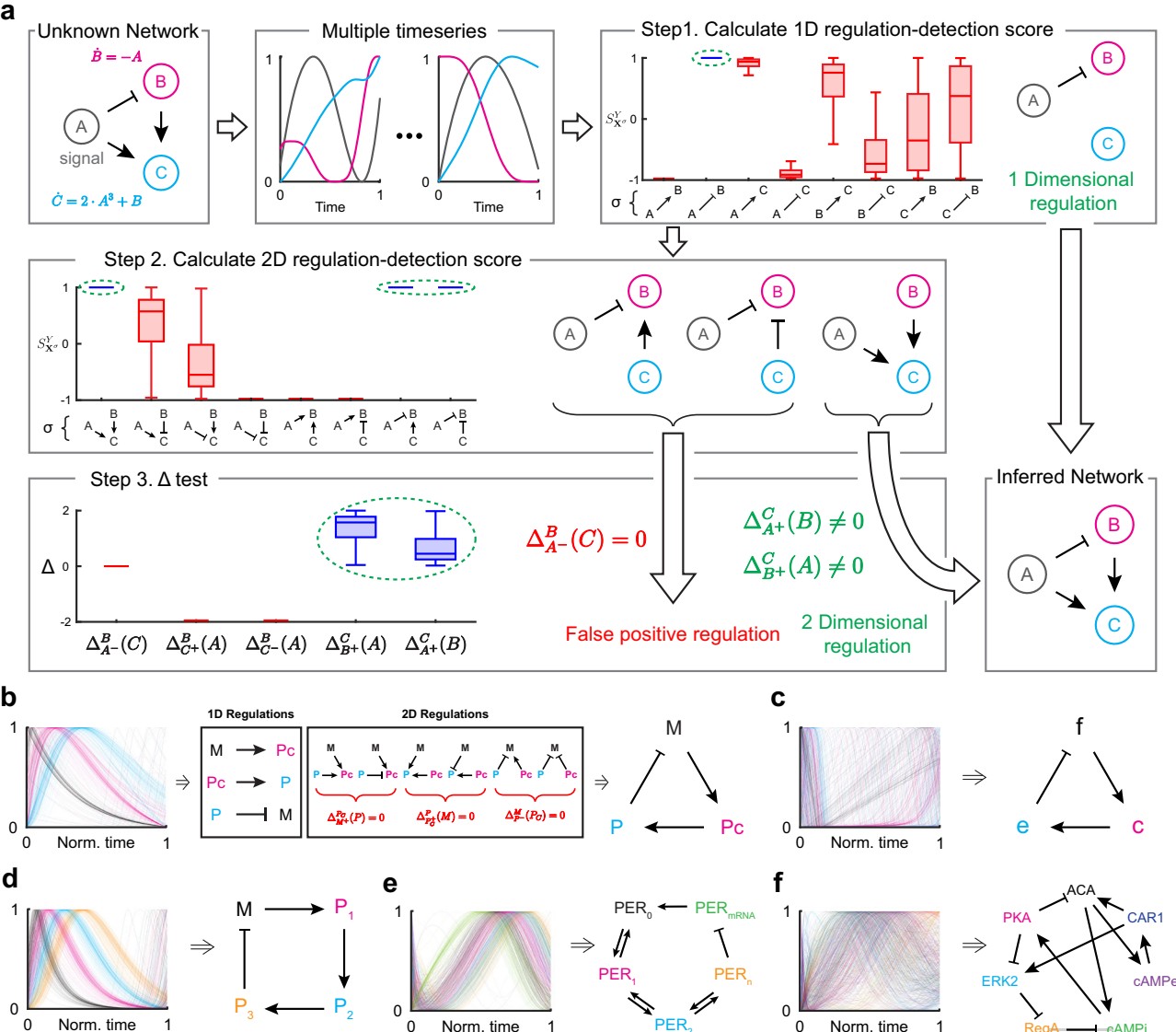

**Fig. 2 | Framework for inferring regulatory networks. a** With ODE describing the network (left), various time series are simulated with different initial conditions (middle). Then, from each time series, the regulation-detection score $S_{\mathbf{X}^\sigma}^Y$ is calculated for every 1D regulation type $\sigma$ (Step 1). The criteria $S_{\mathbf{X}^\sigma}^Y = 1$ infers $A \dashv B$. Next, $S_{\mathbf{X}^\sigma}^Y$ is calculated for every 2D regulation type $\sigma$ (Step 2). Among the three types of regulations with $S_{\mathbf{X}^\sigma}^Y = 1$, only one passes the $\Delta$ test (Step 3). By merging the inferred 2D regulation with the 1D regulation from Step 1, the regulatory network is successfully inferred. Here, data are presented as box plots ($n = 100$), in which the box bounds the IQR divided by the median, and whiskers extend to a maximum of 1.5× IQR beyond the box. **b–f** This framework successfully infers the network structures of the Kim–Forger model (**b**), Frzilator (**c**), the 4-state Goodwin oscillator (**d**), the Goldbeter model for the *Drosophila* circadian clock (**e**), and the cAMP oscillator of *Dictyostelium* (**f**). For each model, 100 time-series data are simulated from randomly selected initial conditions, which lie in the range of the original limit cycle. Source data are provided as a Source Data file.

which successfully distinguishes between the true and false regulations of IFL (Fig. 3c right) and in silico systems investigated in the previous section (Supplementary Fig. 5f). See Methods and Supplementary Information for how to quantify the noise level. Note that $\mathrm{TRS}_{\mathbf{X}^\sigma}^Y$ is the measure that integrates the weight given on the regulation–detection score reflecting the size of the domain of the regulation-detection function (see Supplementary Information for details).

Next, we investigate whether the $\Delta$ test can distinguish direct and indirect regulations using examples of the coherent feed-forward loop (CFL, Fig. 3d) and a single feed-forward loop (SFL, Fig. 3e). In CFL, direct negative regulation from $A$ to $C$ exists. On the other hand, in SFL, only indirect negative regulation from $A$ to $C$, induced from a regulatory chain $A \dashv B \to C$, exists.

In the presence of noise, the regulation-delta function often fails to distinguish these direct and indirect regulations from $A$ to $C$ in CFL and SFL. Specifically, for both CFL and SFL with 20% multiplicative noise, $S_{A^- B^+}^C$ is larger than $S^{\mathrm{thres}}$ and $\Delta_{B^+}^C(A)$ is strictly negative (Fig. 3f, g) for most of the cases. Here, the sign of $\Delta$ is quantified by using a one-tailed Wilcoxon signed rank test (Supplementary Fig. 6a). Thus, the regulation $\begin{smallmatrix} A \dashv \\ B \to \end{smallmatrix} C$ is inferred not only from CFL but also from SFL. This indicates that in the presence of noise, the regulation-delta function can be skewed to the specific type of regulation, even for indirect regulation. To prevent such false positive predictions, we develop another criterion. Specifically, we use a surrogate time series of $A$ ($A_{\mathrm{shuffled}}$, Fig. 3h) to destroy the dependence of $C$ on $A$ in the presence of direct

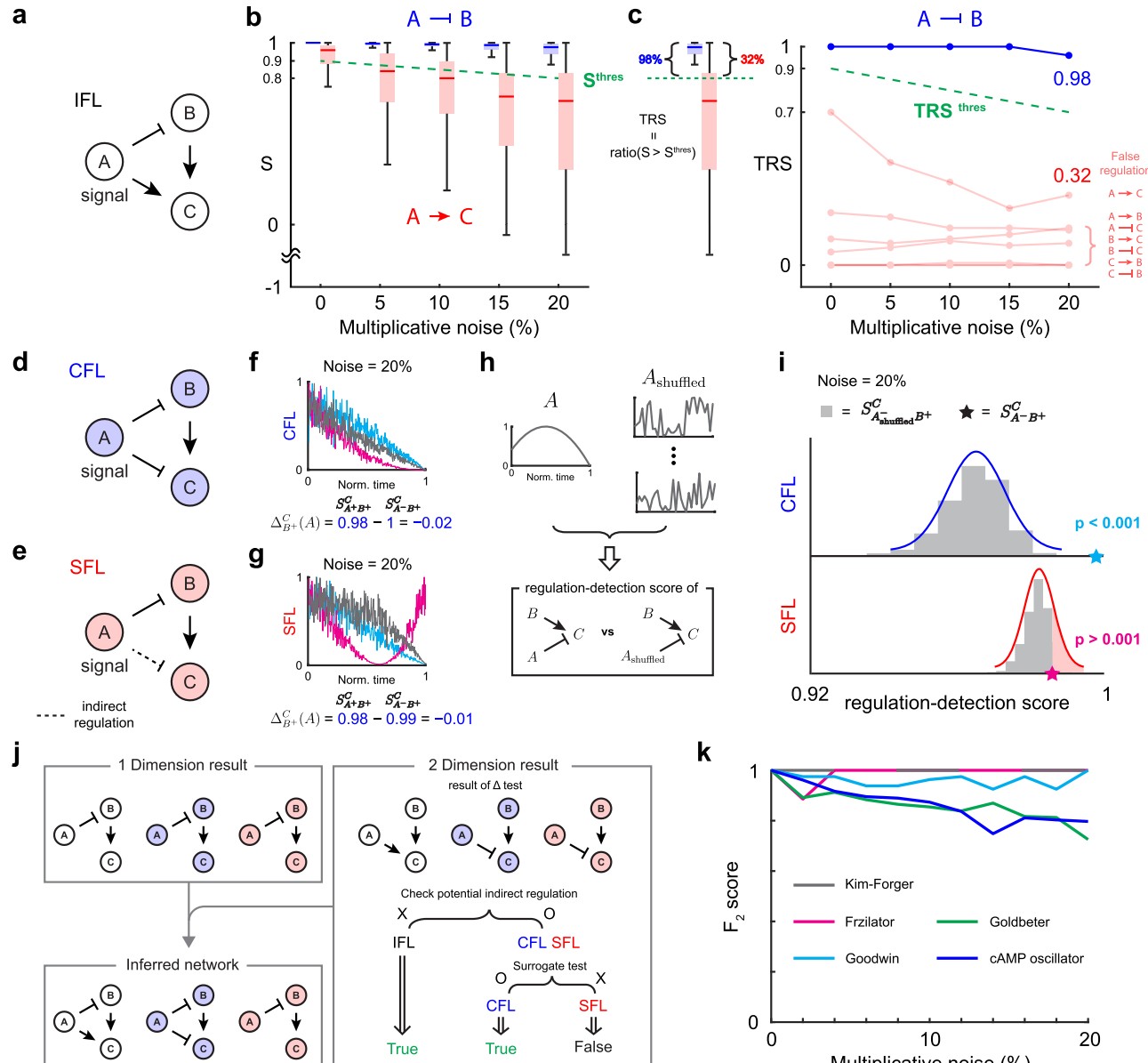

**Fig. 3 | Extended framework for inferring a regulatory network from noisy data.**
**a** A regulatory network with 1D regulation from $A$ to $B$ and 2D regulation from $A$ and $B$ to $C$. **b** The threshold for the regulation-detection score ($S^{thres} = 0.9 - 0.005 \times$ (noise level), green dashed line) distinguishes true ($A \dashv B$) and false regulation ($A \to C$). **c** The fraction of data satisfying $S^\gamma_{X^\sigma} > S^{thres}$, total regulation score ($TRS^\gamma_{X^\sigma}$), is used to infer the regulation. Specifically, $TRS^\gamma_{X^\sigma} > TRS^{thres}$ is used where $TRS^{thres} = 0.9 - 0.01 \times$ (noise level) (green dashed line). Here, data are presented as box plots ($n = 100$), in which the box bounds the IQR divided by the median, and whiskers extend to a maximum of $1.5 \times IQR$ beyond the box. **d** In CFL, direct negative regulation from $A$ to $C$ exists. **e** On the other hand, in SFL, the regulatory chain $A \dashv B \to C$ induces an indirect negative regulation from $A$ to $C$. **f, g** $\Delta^C_{B^+}(A)$ cannot distinguish between the direct and indirect regulations in the

presence of noise because $S^C_{A^- B^+} > S^C_{A^+ B^+}$ for both CFL and SFL, indicating the presence of regulation $A \dashv$ $B \to$ $C$. **h, i** $S^C_{A^-_{shuffled}B^+}$ with the surrogate time series of $A$ can be used to distinguish between the indirect and direct regulations. To disrupt the information of $A$, the time series of $A$ is shuffled (**h**). In the presence of direct regulation (CFL), but not indirect regulation (SFL), $S^C_{A^-_{shuffled}B^+}$ is significantly smaller than the original $S^C_{A^- B^+}$ ($p$-value < 0.001). **j** By including the surrogate test, our extended framework can successfully infer IFL, CFL, and SFL even from noisy time series. **k** $F_2$ score of our inference method when the level of noise increases from 0 to 20%. Here, the mean of the $F_2$ score for 10 data sets is calculated. Each data set consists of 100 time series, which are simulated with different initial conditions. Source data are provided as a Source Data file.

regulation ($A \dashv C$). As a result, the regulation–detection score $S^C_{A^-_{shuffled}B^+}$ is significantly reduced compared to $S^C_{A^- B^+}$ (Fig. 3i top). On the other hand, if $A$ does not directly regulate $C$, then the regulation-detection score $S^C_{A^-_{shuffled}B^+}$ does not decrease much (Fig. 3i bottom), and $S^C_{A^- B^+}$ is not significantly larger than $S^C_{A^-_{shuffled}B^+}$. When multiple time series are given, we calculate the $p$-values for each data and integrate them using Fisher's method. The criteria

(the combined $p$-value < combining $p = 0.001$ for every data) successfully distinguishes between direct and indirect regulation even when the noise level varies (Supplementary Fig. 6b).

From the noisy time series, using the criteria $TRS^\gamma_{X^\sigma} > TRS^{thres}$, all potential 1D (Fig. 3h upper-left) and 2D (Fig. 3h upper-right) regulations are inferred. Then, among the inferred regulations, we need to identify indirect regulations. Unlike IFL, CFL and SFL have a potential indirect regulation. That is, $A \dashv C$ has the potential to be indirect since

there is a regulatory chain $A \dashv B \to C$. In this case, we use a surrogate time series of a potential source of indirect regulation ($A$) to test whether $S^C_{A^- B^+}$ is significantly larger than $S^C_{A_{\text{shuffled}} B^+}$. This reveals that $A \dashv C$ is a direct regulation for CFL, but not SFL. Then, merging 1D and 2D results successfully recovers the network structure of IFL, CFL, and SFL even from noisy time series (Fig. 3j). Since our method involves multi-dimensional inferences, in the presence of noise, various dimensional regulations for a single target can be detected. In this case, only the regulation with the highest value of TRS is inferred. In the example of CFL, our 1D framework infers $B \to C$ and 2D framework infers $\begin{smallmatrix} A \dashv \\ B \to \end{smallmatrix} C$. Since $\text{TRS}^C_{A^- B^+}$ is higher than $\text{TRS}^C_{B^+}$, only 2D regulation $\begin{smallmatrix} A \dashv \\ B \to \end{smallmatrix} C$ is inferred (Fig. 3j).

Based on TRS and post-filtering tests (Δ test and surrogate test), we develop a user-friendly computational package, GOBI, which can be used to infer regulations for systems described by Eq. (1) (see README file on Github[45] and Supplementary Information for manuals). GOBI successfully infers regulatory networks from simulated time series using ODE models (Fig. 2b–f) in the presence of multiplicative noise (Fig. 3k) and other types of noise (Supplementary Fig. 7a). Here, the $F_2$ score, the weighted harmonic mean of precision and recall, is nearly one, indicating that GOBI is able to recover all regulations almost perfectly. However, it should be noted that noise types that significantly affect the shapes of trajectories can result in the decreased performance of GOBI, which uses time series shape information for inference (Supplementary Fig. 7b).

## Successful network inferences from experimentally measured time series

When the proposed thresholds for the regulation-detection score (Fig. 3b) and Total Regulation Score (Fig. 3c) and two critical values of significance (i.e., $p$-value = 0.01 for the Δ test and $p$-value = 0.001 for the surrogate test) are used, GOBI successfully infers the regulatory networks from in silico time series. Here, we use GOBI with these default hyperparameters to infer regulatory networks from experimentally measured time series. From the population data of two unicellular ciliates *Paramecium aurelia* (P) and *Didinium nasutum* (D)[3,46] (Fig. 4a left), the network between the prey (P) and predator (D) is successfully inferred (Fig. 4a and Supplementary Fig. 9a).

Next, we apply GOBI to the time series of the synthetic genetic oscillator, which consists of *Tetracycline repressor* (TetR) and RNA polymerase sigma factor ($\sigma^{28}$)[47] (Fig. 4b left). While the time series are measured under different conditions after adding purified TetR or inactivating intrinsic TetR, our method consistently infers the negative feedback loop, including negative self-regulation based on two direct regulations $\sigma^{28} \to$ TetR and TetR $\dashv \sigma^{28}$ for all cases (Fig. 4b middle and Supplementary Fig. 9b). This indicates that our method can infer regulations even when the data are achieved from different conditions since we do not specify the specific equations with parameters in Eq. (1).

We next investigate the time-series data from a slightly more complex synthetic oscillator, the three-gene repressilator[48] (Fig. 4c left). As the amount of data is greatly reduced compared to the synthetic genetic oscillator (Fig. 4b), we assume negative self-regulation. Then, the criteria $\text{TRS}^Y_{X^\sigma Y^-} > \text{TRS}^{\text{thres}}$ infers three negative 1D regulations and three 2D regulations (Fig. 4c middle). Among the 2D regulations, positive regulations are inferred as indirect as they do not pass the surrogate test (Fig. 4c middle, dashed arrow). Thus, among the inferred 2D regulations, only the negative regulations, consistent with the inferred 1D regulations, are inferred as direct regulations. Gathering these results, GOBI successfully infers the network structure of the repressilator (Fig. 4c right and Supplementary Fig. 9c). Note that although our method infers the regulations among proteins as direct, in fact, mRNA exists as an intermediate step between the negative regulations among the proteins. This happens due to the short translation time in

*Escherichia coli*[49], which causes the mRNA and protein profiles to exhibit similar shapes and phases. This indicates that our method infers indirect regulations with a short intermediate step as direct regulations. Furthermore, compared to the synthetic genetic oscillator (Fig. 4b), the amount of data is small, and the number of components is large; thus, it is essential to assume negative self-regulation for correct inference, i.e., without the assumption, the available data is insufficient to fill the space of the regulation–detection function, making it difficult to detect 2D regulations.

We apply GOBI to the time series measuring the amounts of four cofactors present at the estrogen-sensitive pS2 promoter after treatment with estradiol[50,51] (Fig. 4d left). As all components are expected to decay in proportion to their own concentrations, negative self-regulations are assumed, which is critical due to the small amount of data. GOBI infers five 1D regulations (HDAC ⊣ hER, TRIP1 ⊣ hER, hER → POLII, TRIP1 ⊣ POLII, and HDAC ⊣ POLII) that satisfy the criteria $\text{TRS}^Y_{X^\sigma Y^-} > \text{TRS}^{\text{thres}}$. However, we exclude them because hER and POLII have two and three causes, forming 2D and 3D regulations, respectively, although the 1D criteria assumes a single cause (Fig. 4d middle, dashed box). If all of these regulations are effective, they will be identified as 2D and 3D regulations. Indeed, among the 11 candidates for 2D regulations, most of them include the five inferred 1D regulations. Via Δ test and surrogate test, indirect regulations are identified among inferred 2D regulations (Supplementary Fig. 9d). For example, 2D regulation $\begin{smallmatrix} \text{hER} \to \\ \text{HDAC} \dashv \end{smallmatrix}$ POLII satisfies the criteria $\text{TRS}^Y_{X^\sigma Y^-} > \text{TRS}^{\text{thres}}$. Among two causal variables (i.e., hER and HDAC), only positive regulation from hER passes the post-filtering test, i.e., only 1D regulation hER → POLII, but not HDAC ⊣ POLII is inferred as a direct regulation. Consequently, after excluding all the indirect regulations, two 1D regulations (hER → POLII and HDAC ⊣ hER) and one 2D regulation ($\begin{smallmatrix} \text{POLII} \to \\ \text{TRIP1} \to \end{smallmatrix}$ HDAC) are inferred (Supplementary Fig. 9d). While we are not able to further infer 3D regulations due to the limited amount of data, the inferred regulations are supported by the experiments. That is, estradiol triggers the binding of hER to the pS2 promoter to recruit POLII[50], supporting hER → POLII. Also, inhibition of POLII phosphorylation blocks the recruitment of HDAC but does not affect the APIS engagement at the pS2 promoter[50], supporting POLII → HDAC and no regulation from POLII to TRIP1, which is a surrogate measure of APIS. Without inhibition of POLII, HDAC is recruited after the APIS engagement, and when the HDAC has maximum occupation, then the pS2 promoter becomes refractory to hER[50], supporting TRIP1 → HDAC ⊣ hER. Interestingly, the inferred network contains a negative feedback loop, which is required to generate sustained oscillations[52].

Finally, we investigate five-time series of air pollutants and cardiovascular disease occurrence in Hong Kong from 1994 to 1997[53] (Fig. 4e left). Since our goal is to identify which pollutants cause cardiovascular disease, we fix the disease as a target. Also, we assume the negative self-regulation of disease, reflecting death. While two positive causal links from $NO_2$ and respirable suspended particulates (Rspar) to the disease are identified as 1D regulations (Fig. 4e middle), we exclude them because they share the same target (Fig. 4e middle, dashed box). Among two inferred 2D regulations, one passes the Δ test and surrogate test (Fig. 4d middle). Furthermore, no 3D and 4D regulation is inferred (Supplementary Fig. 9e). The inferred network indicates that both $NO_2$ and Rspar are major causes of cardiovascular diseases (Fig. 4e right). Indeed, it was reported that $NO_2$ and Rspar are associated with hospital admissions and mortality due to cardiovascular disease, respectively[54].

## Comparison between our framework and other model-free inference methods

Here, we compare our framework with popular model-free methods, i.e., GC, CCM, and PCM, by using the experimental time-series data in

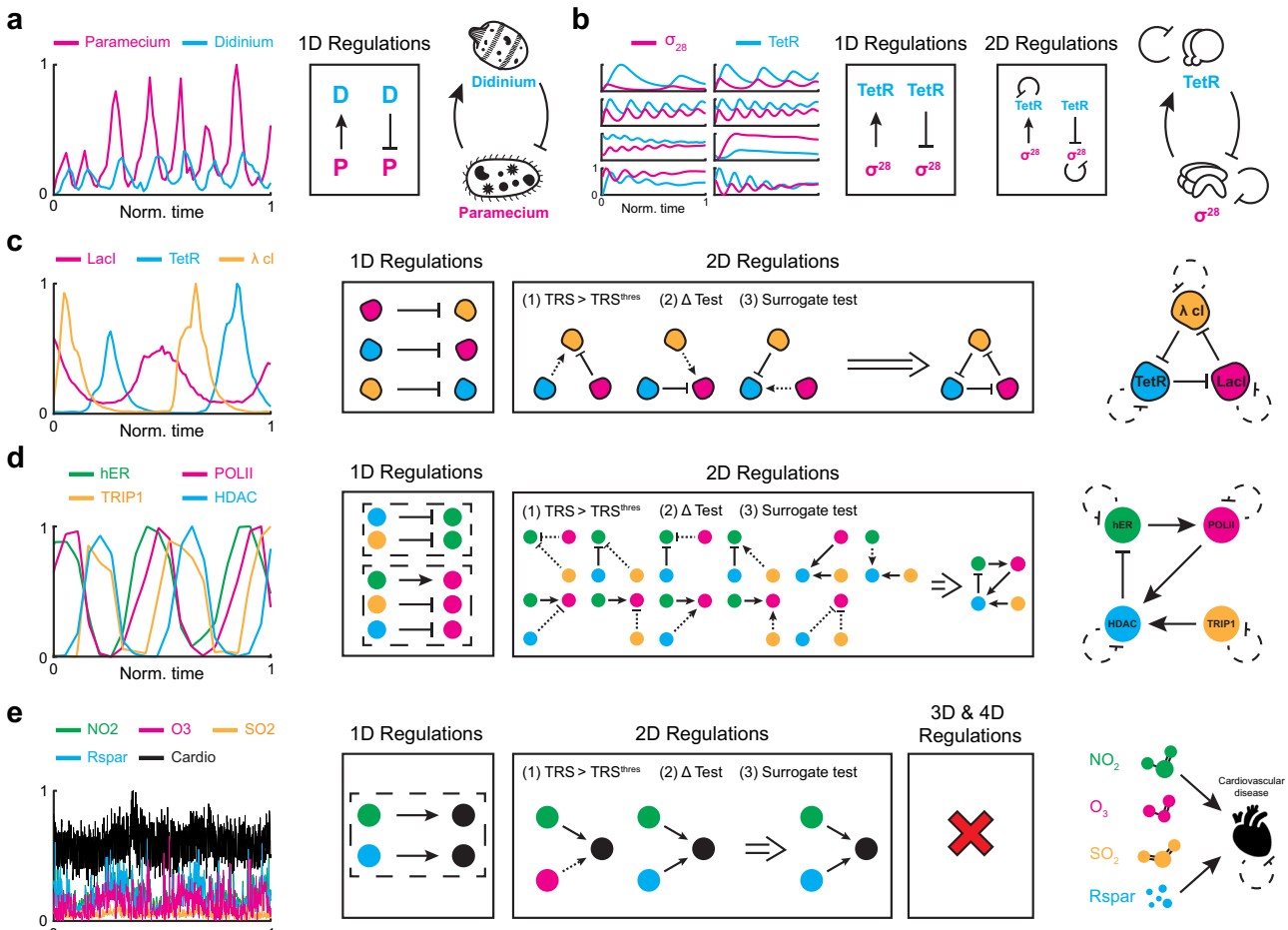

**Fig. 4 | Inferring regulatory networks from experimental data. a** GOBI successfully infers predatory interaction from a 30-day abundance time-series data of two unicellular ciliates *Paramecium aurelia* and *Didinium nasutum* (data is taken from refs. 3, 46). **b** GOBI successfully infers the negative feedback loop including negative self-regulations of the synthetic genetic oscillator consisting of a repressor TetR and activator $\sigma^{28}$ (data is taken from ref. 47). **c** From time-series data of a three-gene repressilator (data is taken from ref. 48), GOBI successfully infers the underlying network. Three direct negative 1D regulations are inferred. Among the three 2D regulations having high TRS, only negative regulations pass the Δ test

and surrogate test. **d** From time series measuring the number of cofactors present at the estrogen-sensitive pS2 promoter after treatment with estradiol (data is taken from ref. 50), five 1D regulations have high TRS. However, they are not inferred because they share a common target (dashed box). Among 11 regulations having high TRS, one 2D regulation and two 1D regulations are inferred, passing the Δ test and surrogate test. **e** From 1000-day time-series data of daily air pollutants and cardiovascular disease occurrence in the city of Hong Kong (data is taken from ref. 20), GOBI finds direct positive causal links from $NO_2$ and Rspar to the disease. Source data are provided as a Source Data file.

the previous section (Fig. 4a–e). Unlike our method, the model-free methods can only infer the presence of regulation and not its type (i.e., positive and negative). Thus, the arrows represent inferred regulations, which could be either positive or negative.

For the prey–predator system and the genetic oscillator (Fig. 4a, b), we merge them to create a more challenging case. Specifically, from the set of eight different time-series data of a genetic oscillator measured under different conditions, we select one that has a similar phase to the time series of the prey–predator system (Fig. 4b panel at the 2nd row and 2nd column). Then, we merge the selected time series with the time series of the prey–predator system. While GOBI and PCM successfully detect two independent feedback loops (Fig. 5a), CCM and GC infer false positive predictions (e.g., P to $\sigma^{28}$ in Fig. 5a) because they usually misidentify synchrony as causality. Furthermore, when we reduce the sampling rate by half, the accuracy of PCM dramatically drops, whereas GOBI can still infer the true network structure (Supplementary Fig. 10).

For a similar reason, synchrony obscures the inference of the model-free methods for the repressilator (Fig. 5b). Moreover, the model-free methods fail to distinguish between direct and indirect regulations. For example, they infer the indirect regulation TetR → $\lambda$cl

induced by the regulatory chain TetR ⊣ LacI ⊣ $\lambda$cl, unlike our method. Similarly, due to synchrony and indirect effect, for the system of cofactors at the pS2 promoter, model-free methods infer an almost fully connected causal network, unlike our method (Fig. 5c).

When we use 3 years of data (full-length data) on air pollutants and cardiovascular disease, PCM infers the same structure as GOBI infers, i.e., only $NO_2$ and Rspar cause the disease (Fig. 5d gray)[20]. On the other hand, when a subset of the data (i.e., two years of data) is used, only GOBI infers the same structure (Fig. 5d purple). This indicates that GOBI is more reliable and accurate than the model-free methods.

## Discussion
We develop an inference method that considerably resolves the weakness of model-free and model-based inference methods. We derive the conditions for interactions satisfying the general monotonic ODE (Eq. (1)). As this allows us to easily check the reproducibility of given time-series data with the general monotonic ODE (i.e., the existence of ODE satisfying given time-series data) without fitting, the computational cost is dramatically reduced compared to the previous model-based approaches. Importantly, as our method can be applied to any system described by general monotonic ODE (Eq. (1)), it

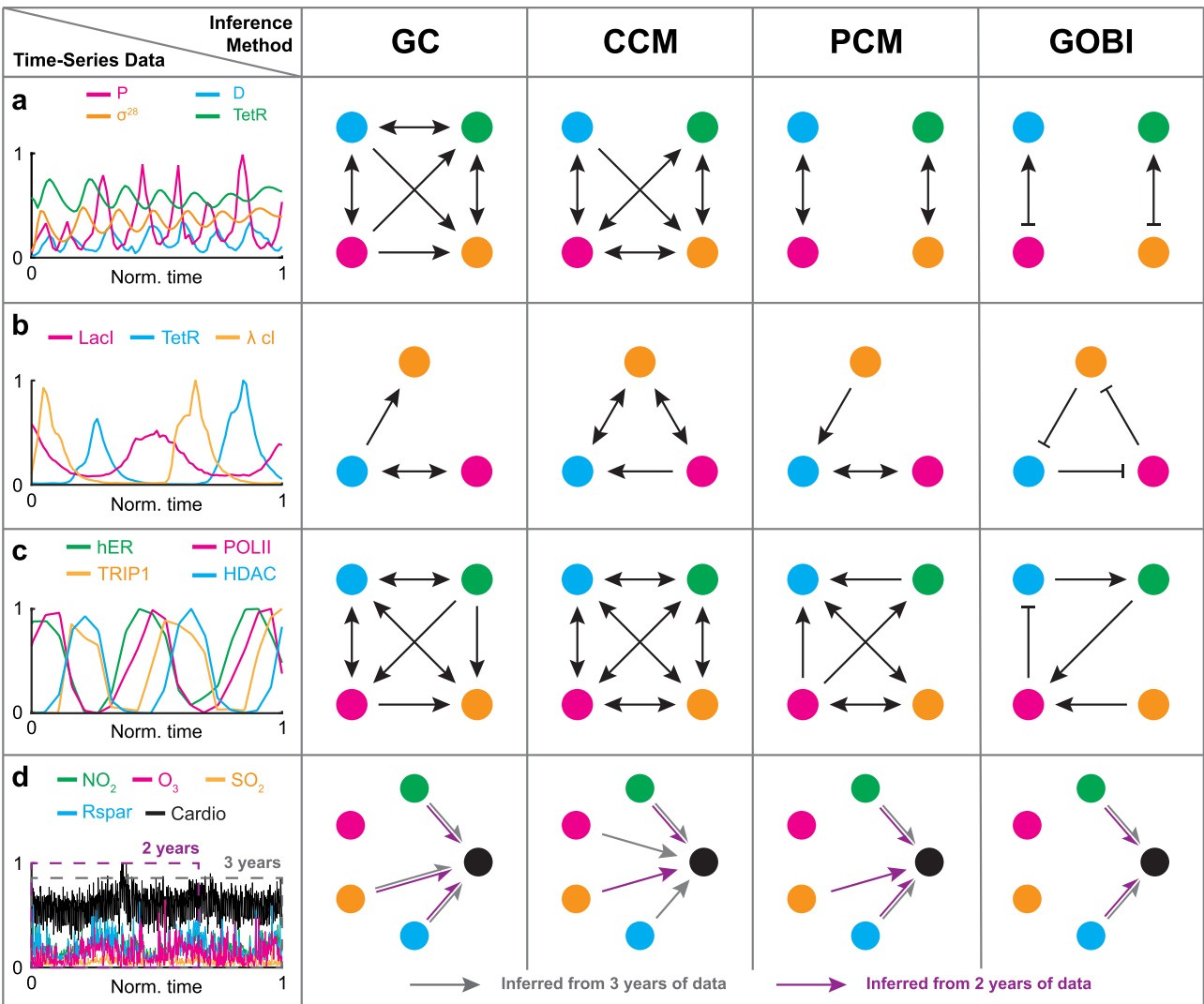

**Fig. 5 | Model-free methods, but not our method, make a false prediction due to the presence of synchrony and indirect effect. a–d** We apply our method and popular model-free methods (i.e., GC, CCM, and PCM) to various experimental time-series data obtained from the prey–predator system merged with the genetic oscillator (**a**); repressilator (**b**); cofactors at the pS2 promoter (**c**); and air pollutants and cardiovascular disease (**d**). For the air pollutants and cardiovascular disease data, we test the methods on three years of data (**d** gray) and on two years of data (**d** purple). Source data are provided as a Source Data file.

significantly addresses the fundamental limit of the model-based approach (i.e., the requirement of a priori model accurately describing the system) (Supplementary Fig. 4). In addition, our method also does not run the serious risk of misidentifying generalized synchrony as causality, unlike the previous model-free approaches. Please note that our approach still cannot deal with completely synchronized system. Furthermore, our method successfully distinguishes direct causal relations from indirect causal relations by adopting the surrogate test (Fig. 3). In this way, our framework dramatically reduces the false positive predictions, which are the inherent flaw of the model-free inference method (Fig. 5). Taken together, we develop an accurate and broadly applicable inference method that can uncover unknown functional relationships underlying the system from their output time-series data (Fig. 4).

Despite these advantages, our method has some limitations that should be addressed. First, our framework assumes that when $X$ causes $Y$, $X$ causes $Y$ either positively or negatively. Thus, GOBI cannot capture the regulation when $X$ causes $Y$ both positively and negatively or when the type of regulation changes over time. However, GOBI can be potentially extended to detect temporal-structured models, including non-monotonic regulation (Supplementary Fig. 11). It would be

interesting in future work to investigate the extended framework thoroughly under diverse circumstances. Additionally, while we have considered the general form of monotonic ODE (Eq. (1)), GOBI can also be extended to describe interactions, including time delays (Supplementary Fig. 12). This will be an interesting future direction to make GOBI more broadly applicable. Also another limitation is the possibility of false positive predictions. This occurs because our method tests the reproducibility of time-series data using necessary conditions. Specifically, the regulation-detection score can be one even in the absence of regulation. To resolve this, we use multiple time-series data and perform post-filtering tests (i.e., Δ test and surrogate test). Nonetheless, it should be noted that inferring high-dimensional regulations requires a large amount of data (Supplementary Fig. 13). To address this challenge, we can use prior knowledge about the system. For example, in biological systems, negative self-regulation can be assumed as the degradation rates of molecules increase as their concentrations increase. By assuming negative self-regulation, we are able to reduce the $N$D regulation to $(N-1)$D regulation, which allows us to successfully infer the network structure even with a small amount of experimental data (Fig. 4c). Note that when a priori assumption (e.g., the types of self-regulation) is not met, only the links that violate the

assumptions are not trustable, i.e., the other inference results are not affected (Supplementary Fig. 3).

To use GOBI, we need to choose hyper-parameters. When applying GOBI to noisy data, users must choose thresholds for the regulation-detection region, regulation-detection function, and total regulation score, as well as two critical values of significance (i.e., $p$-values for $\Delta$ test and surrogate test). In this study, we determine these values by using noisy simulated data of various examples (Fig. 3 and Supplementary Fig. 5). Nevertheless, these values are effective when they are applied to experimental time-series data (Figs. 4 and 5). Thus, we have set those values of hyper-parameters as the default values of GOBI. However, the optimal threshold may vary depending on the data characteristics, and users may need to adjust the thresholds based on the importance of avoiding false positive or false negative predictions. Another hyper-parameter that requires consideration is the choice of the sampling rate. In this study, we use a sampling rate of 100 points per period after evaluating the trade-off between computational cost and accuracy. However, users can decrease or increase the sampling rate if the computation speed is too slow or if a higher level of accuracy is required, respectively.

## Methods

### Computational package for inferring regulatory network

Here, we describe the key steps of our computational package, GOBI (https://github.com/Mathbiomed/GOBI)[45]. For the experimental time-series data $\mathbf{X}(t) = (X_1(t), X_2(t), \cdots, X_N(t))$, $\mathbf{X}(t)$ can be interpolated with either the 'spline' or 'fourier' method, chosen by the user. For the spline interpolation, we use the MATLAB function 'interp1' with the option 'spline', and for the Fourier interpolation, we use the MATLAB function 'fit' with the option 'fourier1–8'. After the interpolation, the derivative of $\mathbf{X}(t)$ is computed using the MATLAB function 'gradient' to compute the regulation–detection score.

### Regulation–detection region.

For the $N$D regulation (Eq. (1)) with regulation type $\sigma$, the regulation-detection region ($R_{\mathbf{X}^\sigma}$) is defined as the set of $(t, t^*)$ on the domain of time series $[0, \tau]^2$ satisfying $\sigma(i)X_i^d(t, t^*) > 0$ for all $i = 1, 2, \cdots, N$. For example, with the positive 1D regulation $X \rightarrow Y$ ($\sigma = +$), $R_{X^+}$ is the set of $(t, t^*)$ where $X^d > 0$. For the 2D regulation $\begin{smallmatrix} X_1 \rightarrow \\ X_2 \dashv \end{smallmatrix} Y$ ($\sigma = (+, -)$), $R_{X_1^+ X_2^-}$ is the set of $(t, t^*)$ satisfying both $X_1^d > 0$ and $X_2^d < 0$. The size of the regulation-detection region (size($R_{\mathbf{X}^\sigma}$)) is the fraction of $R_{\mathbf{X}^\sigma}$ over the domain $[0, \tau]^2$. In the presence of noise, we only consider a region which is not small (i.e., size($R_{\mathbf{X}^\sigma}$) > $R^{thres}$) to avoid an error from the noise. The value of $R^{thres}$ can be chosen from 0 to 0.1, and the choice of $R^{thres}$ does not significantly affect the results (Supplementary Fig. 5a). However, a small value of $R^{thres}$ is recommended for inferring high-dimensional regulations since the average of size($R_{\mathbf{X}^\sigma}$) decreases exponentially as dimension increases (see Supplementary Information for details).

### Regulation–detection function and score.

When the regulation type $\sigma$ from $\mathbf{X} = (X_1, X_2, \cdots, X_N)$ to $Y$ exists, the following regulation-detection function ($I_{\mathbf{X}^\sigma}^Y$) defined on regulation–detection region $R_{X^\sigma}$ is always positive.

$$I_{\mathbf{X}^\sigma}^Y := \dot{Y}^d \cdot \prod_{i=1}^{N} \sigma(i) X_i^d.$$

Thus, the following regulation-detection score ($S_{\mathbf{X}^\sigma}^Y$) is one:

$$S_{\mathbf{X}^\sigma}^Y := \frac{\iint_{R_{\mathbf{X}^\sigma}} I_{\mathbf{X}^\sigma}^Y(t, t^*) \, dt \, dt^*}{\iint_{R_{\mathbf{X}^\sigma}} |I_{\mathbf{X}^\sigma}^Y(t, t^*)| \, dt \, dt^*} \tag{4}$$

(see Supplementary Information for details). However, this is not true anymore in the presence of noise. Thus, we relax the criteria from $S_{\mathbf{X}^\sigma}^Y = 1$ to $S_{\mathbf{X}^\sigma}^Y > S^{thres}$. Among the data which has nonempty $R_{\mathbf{X}^\sigma}$ (i.e., $R_{\mathbf{X}^\sigma} > R^{thres}$), the fraction of data satisfying the criteria $S_{\mathbf{X}^\sigma}^Y > S^{thres}$ is called the total regulation score (TRS$_{\mathbf{X}^\sigma}^Y$). Finally, we infer the regulation from noisy time-series data using the criteria TRS$_{\mathbf{X}^\sigma}^Y$ > TRS$^{thres}$ for noisy time-series data. $S^{thres} = 0.9 - 0.005 \times (\text{noise level})$ and TRS$^{thres} = 0.9 - 0.01 \times (\text{noise level})$ are used (Fig. 3a–c and Supplementary Fig. 5). The noise level of the time series is approximated using the mean square of the residual between the noisy and fitted time series (Supplementary Fig. 8).

### $\Delta$ test.

When we add any regulation to existing true regulation, the regulation-detection score is always one (Fig. 1j-l). Thus, to test whether the additional regulation is effective, we consider $\Delta_{\mathbf{X}^\sigma}^Y(X_{new}) = S_{\mathbf{X}^\sigma X_{new}^+}^Y - S_{\mathbf{X}^\sigma X_{new}^-}^Y$, where $S_{\mathbf{X}^\sigma X_{new}^+}^Y$ ($S_{\mathbf{X}^\sigma X_{new}^-}^Y$) is the regulation-detection score when the new component ($X_{new}$) is positively (negatively) added to the existing regulation type $\sigma$. Because $\Delta_{\mathbf{X}^\sigma}^Y(X_{new}) = 0$ reflects that the new component ($X_{new}$) does not have any regulatory role, the newly added regulation is inferred only when $\Delta_{\mathbf{X}^\sigma}^Y(X_{new}) \neq 0$ for some data. In particular, $\Delta > 0$ ($\Delta < 0$) represents that the new component adds positive (negative) regulation. In the presence of noise, the positive (negative) regulation is inferred if $\Delta \geq 0$ ($\Delta \leq 0$) consistently for all time series. If the number of time series is greater than 25, the sign of $\Delta$ is quantified by a one-tailed Wilcoxon signed rank test. We set the critical value of significance as 0.01, but it can be chosen by the user.

### Surrogate test.

Indirect regulation is induced by the chain of direct regulations. For example, in SFL (Fig. 3e), regulatory chain $A \dashv B \rightarrow C$ induces the indirect negative regulation $A \dashv C$. In the presence of noise, the $\Delta$ test sometimes fails to distinguish between direct and indirect regulations (Fig. 3d–g). Thus, after the $\Delta$ test, if the inferred regulation has the potential to be indirect, we additionally perform the surrogate test to determine whether the inferred regulation is direct or indirect. Specifically, for each candidate of indirect regulation, we shuffle the time series of the cause using the MATLAB function 'perm' and then calculate the regulation–detection scores. Then, we test whether the original regulation-detection score is significantly larger than the shuffled ones by using a one-tailed $Z$ test. In the presence of the $k$ number of time-series data, we can get the $k$ number of $p$-values ($p_i$, $i = 1, 2, \cdots, k$). Thus, we combine them into one test statistic ($\chi^2$) using Fisher's method, $\chi_{2k}^2 \sim -2 \sum_{i=1}^{k} \log(p_i)$. We set the critical value of the significance of Fisher's method by combining $p_i = 0.001$ for all the data, but it can also be chosen by the user.

### Model-free methods.

For CCM[3] and PCM[20], we choose an appropriate embedding dimension using the false nearest neighbor algorithm. Also, we select a time lag producing the first minimum of delayed mutual information. To select the threshold value '$T$' in PCM, we use $k$-means clustering as suggested in[20]. We run CCM using 'skccm' and PCM using the code provided in[20]. For GC[2], we run the code provided in[55], specifying the order of AR processes of the first minimum of delayed mutual information as we choose a max delay with the CCM and PCM. Also, we reject the null hypothesis that $Y$ does not Granger cause $X$, and thereby inferred direct regulations by using the $F$ statistic with a significance level of 95%[2]. Specifically, we use embedding dimension 2 for the prey-predator, genetic oscillator, and estradiol data sets; and 3 for the repressilator and air pollutants and cardiovascular disease data sets. Also, we used time lag 2 for prey–predator; 3–10 for the genetic oscillator (there are eight different time-series data sets); 10 for the repressilator; 15 for the estradiol data set; and 3 for the air pollutants and cardiovascular disease data set.

### in silico time-series data

With the ODE describing the system, we simulate the time-series data using the MATLAB function 'ode45'. The sampling rate is 100 points

per period for all the examples (Figs. 1, 2, and 3). For the multiple time-series data (Figs. 2 and 3), we generate 100 different time series with different initial conditions. Then, before applying our method, we normalize each time series by re-scaling to have minimum 0 and maximum 1. To introduce measurement noise in time series, we introduce multiplicative noise sampled randomly from a normal distribution with mean 0 and standard deviation given by the noise level. For example, for 10% multiplicative noise, we add the noise $X(t_i) \cdot \varepsilon$ to $X(t_i)$, where $\varepsilon \sim N(0, 0.1^2)$. Before applying our method, all the simulated noisy time series are fitted using the MATLAB function 'fourier4'. However, if the noise level is too high, 'fourier4' tends to overfit and capture the noise. Thus, in the presence of a high level of noise, 'fourier2' is recommended for smoothing.

### Experimental time-series data
For the experimental data, we first calculate the period of data by using the first peak of auto-correlation. Then, we cut the time series into periods (Fig. 4a, b). Specifically, we cut the prey-predator time series every five days to generate seven different time series (Fig. 4a). When the number of cycles in the data is low (<5), to generate enough multiple time series (Fig. 4c–e), we cut the data using the moving-window technique. That is, we choose the window whose size is the period of the time series. Then, along the time series, we move the window until the next window overlaps with the current window by 90%. Then, the time series in every window is used for our approach. We employ this approach for the repressilator (Fig. 4c); estradiol data set (Fig. 4d); and air pollution and cardiovascular disease data (Fig. 4e). For instance, we use time-series data of air pollutants and cardiovascular disease with a window size of one year and an overlap of 11 months (i.e., move the window for a month) to generate 23 data sets. Before this, the time series of disease admissions are smoothed using a simple moving average with a window width of seven days to avoid the effect of days of the week. Each time series is interpolated using the MATLAB function 'spline' (Fig. 4a–d) or 'fourier2' (Fig. 4e), depending on the noise level of the time-series data.

### Reporting summary
Further information on research design is available in the Nature Portfolio Reporting Summary linked to this article.

## Data availability
The data sets generated in this study are publicly available on Github[45]. The references for the public data sets used and analyzed during this study can be found in the "Results" section[3,20,46–48,50]. Source data are provided in this paper.

## Code availability
The codes for the GOBI package, including all the figures presented in this article, are publicly available on Github[45].

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

## Acknowledgements

We thank Seokjoo Chae, Hyukpyo Hong, Yun Min Song, and Olive Cawiding for their valuable comments. This work was supported by Samsung Science and Technology Foundation SSTF-BA1902-01 (to J.K.K.) and Institute for Basic Science IBS-R029-C3 (to J.K.K.).

## Author contributions

S.H.P., S.H., and J.K.K. designed the research. S.H.P. and S.H. developed the method. S.H.P. performed computation. S.H.P. analyzed data. J.K.K. supervised the project. All authors wrote the paper.

## Competing interests

The authors declare no competing interests.
