## [Peer Review File · Nature Communications]

A general model-based causal inference method overcomes
the curse of synchrony and indirect effectReviewer #1 (Remarks to the Author):

Using the data as well as the mathematical models, in particular, ODE models, to detect the interaction networks is of paramount significance in the direction of data science, which definitely attracts all the attention from interdisciplinary fields. Certainly, this work falls into this kind of works, containing publishable and interesting results. It provide an algorithm to detect the positive or/and negative interactions among the variables in ODE models. These ODE models could be used to describe a broad range of real-world systems. Due to its novelty and potential interest to the fields, I advocate the publication of the current result; however, if the prestigious NC is the venue for its publication, more justifications are required.

1. Does the current algorithm still need the explicit forms of the ODE vector fields? In real applications, the explicit forms are always incomplete or even unclear. How to deal with this kind of cases? Can some functional bases be used to approximate the vector fields? This is a really important issue. An algorithm, which cannot deal with this issue, will be not that practical in applications.
2. In the manuscript, noise has been taken into account to test the robustness of the developed algorithm. However, it is not clear what is the form of this multiplicative noise and which parameters are incorporated with this kind of noise and why the parameters are selected. Actually, there are many forms of noises, including the additive, the multiplicative noise and colored noise. What about the robustness of the current work to all these noises?
3. It seems that this algorithm's accuracy crucially depends on the sampling rate for the collected experimental data. It is not clear the ceiling/down sampling rate for the accuracy in applications. More justifications need to be done. These justifications are also related to the computational cost of this algorithm.
4. The authors mentioned the possible extension of the current work to the cases containing time delays. However, if the time delays are introduced into the systems, it seems that the current algorithm will not be effective because time delays are unknown, are more complicated, it is hard to locate the detection regions as defined in the current work. I suggest to include a case containing the time delays to show whether the current work is effective or not. Then a much clearer description for the future direction should be provided; otherwise, the impossible extension will also limit the usefulness of this work.
5. Actually, there are a series of methods for parameters estimation for ODE or even PDE models in control theory. Once one has the data, the methods could be directly used to estimate all the parameters and reconstruct the system. Frankly, it sometimes requires high computational cost, but it could be tolerated for the models that are used in the current work. More illustrative examples of high dimensions should be provided to show the possible outstanding advantages of the current work to those model-based methods.
6. Are the monotonic models as investigated in the current work sufficient representative? Are the non- or partially-monotonic models are more prevalent? This justifications also should be further provided. Is there any chance to extend the current work to those nonmontonic or even temporal-structured models?

Reviewer #2 (Remarks to the Author):

The authors attempted to devise a general ODE framework to infer causal relationships between variables from time-series data, with the aim to circumvent limitations of model-free methods on false discoveries and to expedite computation over existing model-based methods that fit various families of specific functions to data so that the underlying mechanistic relationship can be learned. Pivoting on the assumption of monotonic pairwise relationship, the method developed here (GOBI) was shown to detect directed regulation and rule out indirect regulation. The authors tested the efficacy of their method on simulated datasets as well as experimental datasets, claiming that GOBI is accurate and broadly applicable.

Recent years has witnessed a number of endeavors for developing a general ODE framework to infer causal relationship from time-series data (PMID: 27698038, 34675223, 36476735). The theoretical development presented in this study is straightforward but refreshing. If done properly, it could very well be a generic tool for causal inference using time-series data. Unfortunately, the current manuscript has several fundamental issues, raising concerns about the validity and applicability of the proposed method. Below are some general comments.

1. The assumption of monotonicity greatly limits the applicability of GOBI. It is ubiquitous that biological variables interact to influence their dynamics, meaning that monotonic relationship (Equation 1) is extremely rare in real-world scenarios. Since this assumption serves as the foundation for causal inference in GOBI (see Supporting Information Section 1), violations could also confine the validity of the method. Consider a predator-prey model with an interaction term ($Y' = X*Y - X$). Due to the non-monotonicity in X and Y' and the non-independence between X and Y, theorem 1 does not hold and the regulation detection score is bound below 1. This ensures false negative discoveries. Moreover, multivariable monotonicity has not been defined formally.

2. Most examples illustrated in the manuscript have the form of linear combination of single-variable monotonic functions (Fig. 1, Fig. 2a/b/d/e, and Fig. 3). It seems the authors were trying to avoid the aforementioned issues. For the cAMP oscillator (Fig. 2f), where there were interaction terms, the supplementary data indicated that GOBI detected a false regulation (ACA -> cAMPe) and missed a strong prediction (ACA -> cAMPi). These mishaps were omitted in the main figure and not discussed in the main text. Nevertheless, the authors claimed that they recovered the relationship network successfully, prompting necessary doubt over their implementation. Further, the part of inference with noisy data (Fig. 3) was not included in the code, making it impossible to verify or replicate.

3. The manuscript also contains multiple cases of inconsistency and contrived arguments, severely reducing readers' confidence in the contents. For instance, on page 5, the authors mentioned that noise made it difficult for the regulation-detection score to distinguish direct and indirect regulations, thus they needed another criterion for inference, i.e., the surrogate test and combining the p-values. Occasions like this leave one to ponder how limited the method actually is.

4. It is far from clear how GOBI would be a user-friendly tool for the wider community. The analyses on simulated datasets (Fig. 2) appear to involve ad hoc selection that requires domain-specific knowledge or personal discretion (see supplementary data). Inference using experimental data (Fig. 4) was not described in details, and there was no associated supplementary data. It is not entirely obvious how the authors arrived at their findings, other than the schematic summaries. In addition, the code was not annotated for easy adoption and the user's manual is missing.

Specific Comments:

5. Fig. 1a: Notations are misplaced. Zd should be Xd, and X'd should be Y'd.

6. Fig.1d-l: It is not clear what the color bar refers to.

7. Fig. 2a and 3i: Why was the critical value for the Delta test 0.01 and 0.001 for the surrogate test? Were they derived from benchmarking?

8. Fig. 3c: The blue line should be 1 for low noise levels according to Fig. 3b. It is not addressed why the regulation of A -> C was not detected using TRS thres.

9. Fig. 4d-e dashed boxes: Why are regulations not inferred when they share a common target? Is it another limitation of the method?

Reviewer #3 (Remarks to the Author):

The authors propose a new method to detect regulations and their types in a class of systems suited to model biological oscillators. The methodology is applied successfully to both synthetic and experimental data. For this reason, I think it deserves to be published. However, I believe that a revision of the manuscript is necessary.

1. There is a clash between the authors' claim of a general inference method and the details provided in the text, which focuses mainly on computational biology and regulatory networks. For example, in the introduction, model-based inference is presented only focusing on algorithms designed for regulatory networks, and popular and powerful methods such as Kalman Filter (that requires only a model, not restricting it to any class, and can handle both measurement and dynamical noise) are not even mentioned. Interaction inference is a broad field. Methodology and literature are vast and stretch well beyond the papers highlighted by the authors, which are very specific to regulatory networks. The authors should either broaden their view throughout the paper, including applications, or tune it to the designed audience and preferred realm of application (starting from the title).

2. The part of the Discussion section regarding the method limitations is very slim. There are limitations and assumptions here and there (such as thresholds when applying to noisy series) on which the authors should spend few words in the Discussion.

3. The paper has several statements that are just too generous towards the method, such as "our approach completely resolves the fundamental limit of model-based inference: strong dependence on a chosen model." The presented method can only be applied to a specific class of systems, so it sits somewhere between model-based inference and model-free inference. Furthermore, in the noisy case, it depends on two thresholds. Tuning down the text, eliminating these kinds of statements, and highlighting the limitation of the methodology does not take anything out of its value.

4. "In most biological systems, the degradation rates of molecules increase as their own concentrations increase; thus we assume that self-regulation is negative for every component in the system". This is another assumption that was not mentioned in presenting the method. If the method is good, this assumption is not needed. On the contrary, it would be an output of the inference. The authors should show that.

5. What happens in systems where the assumptions are not met? Is all inference messed up? Or only the links violating the assumptions are not trustable?

6. If I well understand, the thresholds for the network inference are derived a posteriori to optimize the inference performance. If so, this should be discussed as a limitation, at least in the Discussion. Which is the sensitivity of the results to the threshold values? And how should the user proceed to choose the thresholds if ground-truth data is not available?

Some additional minor points:

Sec II A: "positive or negative causation" maybe the authors meant "regulation" rather than "causation"

Sec II A: The authors should state clearly how Δ is extended to three and more dimensions. The reader can get the idea but being explicit is always better.

Sec IIC: The noise in the time series is measurement noise, not dynamical, isn't it? It should be stated clearly in the text. Also, it would be interesting to see, at least for a simple negative feedback loop of two nodes, what would be the impact of dynamical noise.

How much data is needed to run the inference? How does it depend on the order of the interaction? Is it suitable for large networks? I understand that the space to compute S goes down exponentially but I think the paper needs more quantitative statements

Since the authors present an implementation, it would be nice to know the order of magnitude of the runtime

In the supplementary material, the description of the noise level calculation could be improved. Just saying that the library uses a MATLAB function (of which I couldn't find the documentation) without mentioning what the function does is a bit short, being that an ingredient of the calculation. Same for the MATLAB function "gradient". Is it applied before or after the noise filtering? I guess after, but it should be specified. And what is the underlying algorithm?

In the calculation of GC, I couldn't find how you chose the order of the AR processes for Y and X.

Speaking of GC, if you shift one series and test GC $X(t) \diamond X(t+T)$ (Fig.4ab) and GC does not detect that as a link, I would be surprised, as the shifted timeseries becomes an AR process of its past, which is exactly what GC tests for. Maybe the authors could discuss more in depth why they chose to employ this test. Alternatively, I think that if they want a null hypothesis for non-connection, it could have more sense to use a series from another dataset (such as TetR & P & D).

We thank the reviewers for their positive feedback on our manuscript, constructive comments, and suggestions. In response, we have heavily revised the manuscript. Below, we give our detailed responses to the comments we received and describe the changes in the manuscript. The comments appear in black, our responses in blue, and the changed parts of our manuscript in *italics*.

Reviewer: 1

Using the data as well as the mathematical models, in particular, ODE models, to detect the interaction networks is of paramount significance in the direction of data science, which definitely attracts all the attention from interdisciplinary fields. Certainly, this work falls into this kind of works, containing publishable and interesting results. It provides an algorithm to detect the positive or/and negative interactions among the variables in ODE models. These ODE models could be used to describe a broad range of real-world systems. Due to its novelty and potential interest to the fields, I advocate the publication of the current result; however, if the prestigious NC is the venue for its publication, more justifications are required.

1. Does the current algorithm still need the explicit forms of the ODE vector fields? In real applications, the explicit forms are always incomplete or even unclear. How to deal with this kind of cases? Can some functional bases be used to approximate the vector fields? This is a really important issue. An algorithm, which cannot deal with this issue, will be not that practical in applications.

Our algorithm, called General ODE-Based Inference (GOBI), does not require explicit knowledge of the ODE vector fields. Specifically, GOBI aims to infer the regulation of Y by multiple components X , described by the general ODE model (Eq. (1)), where f represents any smooth function with increasing or decreasing behavior with respect to X_i . Unlike the majority of previous model-based inference methods, our algorithm (GOBI) does not require a specific form of f , making it highly practical for real applications. Thus, all inferences were performed without making assumptions about the explicit form of the ODE (Fig. 2-4). This is an important aspect of our approach, which we have discussed in detail in the manuscript as follows:

- Section I (page 1): “Here, we develop a model-based method that infers interactions among multiple components described by the general *monotonic* ODE model:

$$\frac{dY}{dt} = f(X) = f(X_1, X_2, \dots, X_N),$$

where f can be any smooth and monotonic increasing or decreasing functions of X_i and X_N is Y in the presence of self-regulation. Thus, our approach *considerably* resolves the fundamental limit of model-based inference: strong dependence on a chosen model.”

- Section III (page 8): "... Importantly, as our method can be applied to any system described by general *monotonic* ODE (Eq. (1)), it *significantly addresses* the fundamental limit of the model-based approach (i.e., requirement of a priori model accurately describing the system). ..."

2. In the manuscript, noise has been taken into account to test the robustness of the developed algorithm. However, it is not clear what is the form of this multiplicative noise and which parameters are incorporated with this kind of noise and why the parameters are selected. Actually, there are many forms of noises, including the additive, the multiplicative noise and colored noise. What about the robustness of the current work to all these noises?

We thank the reviewer for this comment. In our study, we generated the noisy time-series data by introducing multiplicative noise to the time series simulated from ODE, which we discussed in the Methods section as follows:

Section IV B (page 11): "... *To introduce measurement noise in time series, we introduce multiplicative noise sampled randomly from a normal distribution with mean 0 and standard deviation given by the noise level. For example, for 10% multiplicative noise, we add the noise $X(t_i) \cdot \varepsilon$ to $X(t_i)$, where $\varepsilon \sim N(0, 0.1^2)$*"

Furthermore, we have now tested the robustness of GOBI to various types of noises, including additive noise, colored noise, and dynamical noise, and discussed it as follows:

Section II C (page 6): "... GOBI successfully infers regulatory networks from simulated time series using ODE models (Fig. 2b-g) in the presence of *multiplicative noise (Fig. 3k) and other types of noise (Supplementary Fig. 7a)*. Here, the F_2 score, the weighted harmonic mean of precision and recall, is nearly one, *indicating that GOBI is able to recover all regulations almost perfectly. However, it should be noted that noise types that significantly affect the shapes of trajectories can result in the decreased performance of GOBI that uses time-series shape information for inference (Supplementary Fig. 7b)*."

- Supplementary Information:

Section X. The robustness of GOBI to various types of noise

In the main text, we considered multiplicative noise as the measurement noise (Fig. 3). Here, we investigate the robustness of GOBI by introducing different noise models, such as additive noise, colored noise, and dynamical noise.

First, let $X = \{X(t_1), X(t_2), \dots, X(t_N)\}$ denote the original time series that are normalized after being simulated from the ODE. To generate the noisy time series with the multiplicative noise level of $m\%$, we randomly sample a noise $\varepsilon = \{\varepsilon_1, \varepsilon_2, \dots, \varepsilon_N\}$ from a normal distribution with a mean of 0 and a standard deviation of 1 (i.e., $\varepsilon_i \sim N(0, 1)$, where $i = 1, 2, \dots, N$). Then, we multiply the noise ε proportional to the noise level and add it to the original time series to obtain the noisy time series as follows:

$$X_{\text{multiplicative noise}} := \{X(t_1) \cdot (1 + \frac{m}{100} \cdot \varepsilon_1), X(t_2) \cdot (1 + \frac{m}{100} \cdot \varepsilon_2), \dots, X(t_N) \cdot (1 + \frac{m}{100} \cdot \varepsilon_N)\}.$$

Next, we generate noisy time series with additive noise. Following a procedure used for generating time series with multiplicative noise, we randomly sample the noise ε from a normal distribution $N(0, 1)$. Then, we add the noise to the original time series proportional to the noise level as follows:

$$X_{\text{additive noise}} := \{X(t_1) + \frac{m}{100} \cdot \varepsilon_1, X(t_2) + \frac{m}{100} \cdot \varepsilon_2, \dots, X(t_N) + \frac{m}{100} \cdot \varepsilon_N\}.$$

Third, we generate noisy time series with colored noise, utilizing two representative types: pink noise and blue noise. Pink noise is characterized by strong power at low frequency, while blue noise is characterized by strong power at high frequency. We sample pink noise $\varepsilon^{\text{pink}} = \{\varepsilon_1^{\text{pink}}, \varepsilon_2^{\text{pink}}, \dots, \varepsilon_N^{\text{pink}}\}$ and blue noise $\varepsilon^{\text{blue}} = \{\varepsilon_1^{\text{blue}}, \varepsilon_2^{\text{blue}}, \dots, \varepsilon_N^{\text{blue}}\}$ using the MATLAB function 'dsp.ColoredNoise'. Then, we generate noisy time series with a colored noise level of $m\%$ as follows:

$$X_{\text{pink noise}} := \{X(t_1) \cdot (1 + \frac{m}{100} \cdot \varepsilon_1^{\text{pink}}), X(t_2) \cdot (1 + \frac{m}{100} \cdot \varepsilon_2^{\text{pink}}), \dots, X(t_N) \cdot (1 + \frac{m}{100} \cdot \varepsilon_N^{\text{pink}})\} \text{ and}$$

$$X_{\text{blue noise}} := \{X(t_1) \cdot (1 + \frac{m}{100} \cdot \varepsilon_1^{\text{blue}}), X(t_2) \cdot (1 + \frac{m}{100} \cdot \varepsilon_2^{\text{blue}}), \dots, X(t_N) \cdot (1 + \frac{m}{100} \cdot \varepsilon_N^{\text{blue}})\}.$$

Until now, we have only taken into account the effects of measurement noise. To make GOBI more widely applicable, we also need to consider the influence of dynamical noise. To generate noisy time series with dynamical noise, we add a noise term to the ODE. Specifically, let the dynamic of Y be given as follows:

$$\frac{dY}{dt} = f(X).$$

Then, the noisy time series $Y_{\text{dynamical noise}}$ with the dynamical noise level of $m\%$ is simulated from the following ODE:

$$\frac{dY_{\text{dynamical noise}}}{dt} = f(X) + Y_{\text{mean}} \cdot \frac{m}{100} \cdot \varepsilon,$$

where Y_{mean} is the mean of the original time series Y , and ε is randomly sampled from a normal distribution $N(0, 1)$.

Now, we illustrate the robustness of GOBI to different types of noise. Using various ODE models (Fig. 2b-f), we simulate 100 time series from randomly selected initial conditions which lie in the range of the original limit cycle. For each time series, we introduce five types of noise, with noise levels varying from 0 to 20%, as previously described. From each noisy time-series data set, we use the Fourier method (i.e., using the MATLAB function 'fit' with the 'fourier4' option) to obtain fitted time-series data, and then apply GOBI to infer a network structure. Then,

we calculate the F_2 score for each system, noise type, and noise level. This process is repeated ten times, and the mean F_2 scores are presented (Supplementary Fig. 7a).

For simple systems, such as the Kim-Forgor model, Frzillator, and 4-state Goodwin oscillator, GOBI successfully infers regulatory networks regardless of the type of noise in the simulated time-series data (Supplementary Fig. 7a (i)-(iii)). However, for more complex systems that include 2D regulations, such as the Goldbeter model and cAMP oscillator, GOBI exhibits varying levels of robustness under different types of noise (Supplementary Fig. 7a (iv) and (v)). Specifically, the performance of GOBI is greatly affected by additive noise and pink noise compared to other types of noise. Additive noise is added independently of the original signal's value, whereas multiplicative noise is added as a proportion of the signal. Thus, additive noise has a greater impact on the signal's overall shape and magnitude than multiplicative noise (Supplementary Fig. 7b (i) and (ii)). Additionally, pink noise has more power at low frequencies than blue noise, making it more likely to affect the shape of time series (Supplementary Fig. 7b (iii) and (iv)). Taken together, noise types that significantly affect the shape of trajectories, such as additive noise and pink noise, can result in the decreased performance of GOBI because the algorithm uses time series shape information for inference.

Supplementary Fig. 7. The robustness of GOBI to various types of noise. **a** The robustness of GOBI is tested under different types of noise added to time-series data simulated using various ODE models: Kim-Forgor model (**a** (i)), Frzillator (**a** (ii)), 4-state Goodwin oscillator (**a** (iii)), Goldbeter model (**a** (iv)), and cAMP oscillator (**a** (v)). For each time-series data, five different types of noise, including multiplicative noise, additive noise, pink noise, blue noise, and dynamical noise, are added with noise levels varying from 0 to 20%, respectively. For each noisy time-series data set, GOBI infers a network structure and the F_2 score is calculated. This process is repeated ten times and the mean F_2 score is shown for each system, noise type, and noise level (**a** (i)-(v)). Here, each data set consists of 100 time series simulated with different initial conditions. **b** The original time series is simulated from cAMP oscillator (black line), and a

*20% level of noise is added (grey line) for each type of noise: multiplicative noise (**(i)**), additive noise (**(ii)**), pink noise (**(iii)**), blue noise (**(iv)**), and dynamical noise (**(v)**). Then, the Fourier method is applied to fit the noisy time series (blue line).*

3. It seems that this algorithm's accuracy crucially depends on the sampling rate for the collected experimental data. It is not clear the ceiling/down sampling rate for the accuracy in applications. More justifications need to be done. These justifications are also related to the computational cost of this algorithm.

We agree with the reviewer's comment that the accuracy of GOBI depends on the sampling rate of the collected experimental data, because a low sampling rate can result in time series with a different shape compared to data collected at a high sampling rate. To address this concern, we have now tested the accuracy of GOBI on experimental data by adjusting the sampling rates and found that it is less sensitive compared to the model-free methods as described below (Supplementary Fig. 10). We have also discussed this issue in the Discussion section (see Reviewer 3's comment 2). Lastly, we have now included a discussion of the computational cost of GOBI in the Supplementary Information (see Reviewer 3's comment 11).

- Supplementary Information:

Section XIII. The accuracy of GOBI on experimental data at different sampling rates

Here, we investigate how the accuracy of GOBI and model-free methods (GC, CCM, and PCM) is affected by the sampling rates of experimental time-series data obtained from systems with known regulatory networks (Fig. 5a-b). After varying the sampling rates of each time-series data, we apply GOBI and model-free methods, including GC, CCM, and PCM, and compute the F_2 scores of the inference results (Supplementary Fig. 10b). Model-free methods infer a lot of false positive predictions regardless of the sampling rate because they often misidentify synchrony for causality (Supplementary Fig. 10c). While PCM can infer the true network structure (i.e., two independent feedback loops) of the prey-predator system merged with the genetic oscillator from the original experimental data, its accuracy drops dramatically when the sampling rate is reduced (Supplementary Fig. 10b (i)). However, GOBI successfully infers true network structures even when the sampling rate is halved (Supplementary Fig. 10b-c). When the sampling rate is reduced by a quarter, which significantly changes the shape of the time series (Supplementary Fig. 10a (i)-(ii) bottom), the accuracy of GOBI also decreases, but it is still comparable to that of model-free methods (Supplementary Fig. 10b-c). This indicates that GOBI is robust to changes in the sampling rate as long as the shape of the time series is preserved.

Supplementary Fig. 10. Accuracy of GOBI on experimental data at different sampling rates. **a** Time-series data from two different systems are used: the prey-predator system merged with the genetic oscillator (**a** (i)) and the repressilator (**a** (ii)). For each system, time-series data are obtained at different sampling rates: using all time points (**a** (i)-(ii) top), one for every two adjacent time points (**a** (i)-(ii) middle), and one for every four adjacent time points (**a** (i)-(ii) bottom). **b** Using time-series data with varying sampling rates, the accuracy of GOBI and model-free methods (GC, CCM, and PCM) is tested and F_2 scores are calculated. **c** For each system, the inferred network structures at different sampling rates are illustrated.

4. The authors mentioned the possible extension of the current work to the cases containing time delays. However, if the time delays are introduced into the systems, it seems that the current algorithm will not be effective because time delays are unknown, are more complicated, it is hard to locate the detection regions as defined in the current work. I suggest to include a case containing the time delays to show whether the current work is effective or not. Then a much clearer description for the future direction should be provided; otherwise, the impossible extension will also limit the usefulness of this work.

We have now extended GOBI to the case containing time delays. In particular, we have investigated how GOBI can be used to infer time-delayed causal interactions using a simple example taken from [Glass, D.S. et al., Nat Commun, 2021] as described below (Supplementary Fig. 12). We have also discussed this in the Discussion section (see Reviewer 3's comment 2).

• Supplementary Information:

Section XV. Extension to infer time-delayed causal interactions

When X positively regulates Y with time delay τ_X , specifically when the dynamic of Y is given as $\frac{dY(t)}{dt} = f(X(t - \tau_X))$, the positive relationship between $X(t - \tau_X)$ and $\dot{Y}(t)$ can be captured by extending the regulation-detection function with a time delay as follows:

$$I_{X^+}^Y(t, t^*, \tau_X) := X^d(t, t^*, \tau_X) \times \dot{Y}^d(t, t^*) := (X(t - \tau_X) - X(t^* - \tau_X)) \times (\dot{Y}(t) - \dot{Y}(t^*)).$$

In the presence of positive regulation from X to Y with time delay τ_X , $I_{X^+}^Y(t, t^*, \tau_X)$ is always positive where $X^d(t, t^*, \tau_X) > 0$, and its normalized integral (i.e., extended regulation-detection score with a time delay),

$$S_{X^+}^Y(\tau_X) := \frac{\iint_{X^d(t, t^*, \tau_X) > 0} I_{X^+}^Y(t, t^*, \tau_X) dt dt^*}{\iint_{X^d(t, t^*, \tau_X) > 0} |I_{X^+}^Y(t, t^*, \tau_X)| dt dt^*} \quad (S2)$$

is one.

This idea can be applied to the case of delayed negative regulation. In the presence of negative regulation $X \dashv Y$ with time delay τ_X , the extended regulation-detection function with a time delay,

$$I_{X^-}^Y(t, t^*, \tau_X) := -X^d(t, t^*, \tau_X) \times \dot{Y}^d(t, t^*) := -(X(t - \tau_X) - X(t^* - \tau_X)) \times (\dot{Y}(t) - \dot{Y}(t^*))$$

can capture the positive relationship between $-X(t - \tau_X)$ and $\dot{Y}(t)$. Thus, $I_{X^-}^Y(t, t^*, \tau_X)$ is always positive where $-X^d(t, t^*, \tau_X) > 0$, and its normalized integral,

$$S_{X^-}^Y(\tau_X) := \frac{\iint_{-X^d(t, t^*, \tau_X) > 0} I_{X^-}^Y(t, t^*, \tau_X) dt dt^*}{\iint_{-X^d(t, t^*, \tau_X) > 0} |I_{X^-}^Y(t, t^*, \tau_X)| dt dt^*} \quad (S3)$$

is one.

Now, we illustrate how to infer time-delayed causal interactions using a simple example, taken from (Glass, D.S. et al., Nat Commun, 2021), where X negatively regulates Y with time delay $\tau_X = 3$ (Supplementary Fig. 12a). To infer the delayed regulation, we use the extended regulation-detection score with a time delay (Eq. (S2) and (S3)). We compute $S_{X^-}^Y(\tau)$ for each time-series data with different values of τ , varying from 0 to 5 (Supplementary Fig. 12b). $S_{X^-}^Y(\tau)$ is always one only when the τ is equal to τ_X , whereas if τ is not equal to τ_X , $S_{X^-}^Y(\tau)$ is not always one. Thus, GOBI can be used to infer delayed regulation and detect the corresponding time delay. This idea can be extended to cases with multi-dimensional regulation, which would be an interesting future direction.

*Supplementary Fig. 12. Inferring delayed regulation using an extended regulation-detection score. **a** The network motif contains delayed negative regulation from X to Y (**a** top). The network is described by a delay differential equation, with $\tau' = 5$ and $\tau_X = 3$ as time delays (**a** bottom). Various time series are simulated from different initial conditions, specifically $Y(0) = 1$ and $X(t) = \sin \sin \left(t + \frac{\pi}{10}j \right)$ on $t \in [-5, 0]$ where $j = 1, 2, \dots, 20$. **b** $S_X^Y(\tau)$ is computed for each time-series data with different values of τ , varying from 0 to 5. The mean value and standard deviation of $S_X^Y(\tau)$ over 20 time-series data are shown.*

5. Actually, there are a series of methods for parameters estimation for ODE or even PDE models in control theory. Once one has the data, the methods could be directly used to estimate all the parameters and reconstruct the system. Frankly, it sometimes requires high computational cost, but it could be tolerated for the models that are used in the current work. More illustrative examples of high dimensions should be provided to show the possible outstanding advantages of the current work to those model-based methods.

We agree with the reviewer that the computational cost of the model-based methods can be tolerable with the advance in the algorithms and GPU/CPU. However, the major drawback of the model-based methods is that they heavily rely on the selection of a model because choosing an unsuitable model can result in false predictions. We have discussed this in the manuscript as follows:

- Section I: "... Although testing the reproducibility is computationally expensive, as long as the underlying model is accurate, the model-based inference method is accurate even in the presence of synchrony in time series and indirect effect [21-29]. However, the inference results strongly depend on the choice of model, and inaccurate model imposition can result in false positive predictions, limiting their applicability. ..."

We have now demonstrated how GOBI can offer advantages over the model-based approach by providing a simple example as shown below (Supplementary Fig. 4). We have also discussed this issue in the Results section and Discussion section (see Reviewer 3's comment 2).

• Section III: “... Taken together, our method successfully infers regulatory networks from various *in silico* systems regardless of their explicit forms of ODE by assuming a general monotonic ODE (Eq. (1)). Unlike our approach, model-based methods that require specifying the model equations produce inaccurate inferences if inappropriate functional bases are chosen (Supplementary Fig. 4).”

• Supplementary Information:

Section VII. Advantages of GOBI over model-based methods for predicting causal relationships

In this section, we demonstrate how GOBI can be more effective than model-based approaches for predicting causal relationships. Model-based methods have a significant limitation in that their inference results depend on the choice of model. If an inappropriate mathematical model is chosen, it can lead to false predictions. For example, when attempting to model a positive regulation from A to B ($A \rightarrow B$), there are various ways to express this relationship mathematically. If an unsuitable model is chosen, the fitting of the model can inform the wrong causal relationship. However, GOBI does not require explicit knowledge of the underlying mathematical model and is thus free from such assumptions. To illustrate this advantage, we use a simple system with two components A and B (Supplementary Fig. 4a (i)-(ii)). Here, A positively regulates B since $\frac{1}{3}A^3 - A^2 + A + 1$ is an increasing function with respect to A . With the ODE describing the network, we simulate a time-series data with initial condition of $B(0) = 1$ and input signal A (Supplementary Fig. 4a (iii)). The time series of input signal A is constructed by connecting 51 randomly selected points from $N(1, 0.5)$ over the time domain $[0, 50]$ with the spline fitting.

Suppose we do not know the underlying dynamics of this system and use the following hill-type functions to test for the existence of regulation from A to B (Supplementary Fig. 4b (i)), which has been widely used in the previous model-based inference (Gotoh, T. et al., Proceedings of the National Academy of Sciences, 2016; Lillacci, G. & Khammash, M., PLoS computational biology, 2010; Toni, T. et al., Journal of the Royal Society Interface, 2009):

$$\dot{B} = \alpha \frac{A^n}{K_1 + A^n} + \beta \frac{K_2}{K_2 + A^m} - \gamma B.$$

This ODE includes both positive ($\alpha \frac{A^n}{K_1 + A^n}$) and negative ($\beta \frac{K_2}{K_2 + A^m}$) regulation from A to B . To estimate the parameters $\alpha, \beta, \gamma, K_1, K_2, n$, and m , we use simulated annealing with parameter ranges of $\alpha, \beta, \gamma \in [0, 5], K_1, K_2 \in [0, 20],$ and $n, m \in [1, 5]$. We collect 100 parameter sets that reproduced the time-series data (Supplementary Fig. 4b (ii)). The values of the parameters corresponding to positive regulation (α) and negative regulation (β) from A to B are similar, indicating the presence of both positive and negative regulation from A to B (Supplementary Fig. 4b (iii)). Consequently, the model-based method fails to detect the positive regulation from A to B . This example highlights the importance of selecting the correct underlying model to achieve

accurate predictions of causal relationships using model-based methods, which can limit the effectiveness of these approaches.

We then apply GOBI, which does not need to specify underlying regulatory functions, to the same data. First, we employ the moving-window technique to divide the time series into 41 different segments with a length of 10, where each segment overlaps with the previous one by 10%. From each time-series data, we compute the regulation-detection scores for all possible 1D and 2D regulations (Supplementary Fig. 4c (i)-(ii) left). None of the 1D regulations pass the criteria $S_{X^+}^Y = 1$ (Supplementary Fig. 4c (i) right), while one 2D regulation passes the criteria (Supplementary Fig. 4c (ii) right). Also, $\Delta_{A^+}^B(B), \Delta_{B^-}^B(A) \neq 0$ indicating that this 2D regulation is not a false-positive regulation. By merging the results, GOBI successfully recovers the underlying regulatory network. This example illustrates the effectiveness of GOBI in cases where the underlying regulatory functions are unknown, in contrast to model-based methods that may fail under such circumstances.

Supplementary Fig. 4. Comparing model-based method and GOBI for inferring a regulatory network. **a** The ODE (**a (i)**) describes a regulatory network (**a (ii)**) with two components. Here, A positively regulates B (i.e., $\frac{1}{3}A^3 - A^2 + A + 1$ is an increasing function with respect to A). With the ODE describing the network, the time series of B is simulated with initial condition $B(0) = 1$ and the time series of input signal A , which is constructed by connecting randomly selected 51 points from $N(1, 0.5)$ over the time domain $[0, 50]$ using the spline fitting (**a (iii)**). **b** Model-based

method is tested with assuming two different hill functions ($\alpha \frac{A^n}{K_1 + A^n}$ and $\beta \frac{K_2}{K_2 + A^m}$) to describe the positive and negative regulation from A to B (**b (i)**). We use simulated annealing to estimate seven parameters $\alpha, \beta, \gamma, K_1, K_2, n,$ and m . Here, parameters are selected within a range of $\alpha, \beta, \gamma \in [0, 5], K_1, K_2 \in [0, 20],$ and $n, m \in [1, 5]$. For 100 pairs of parameters that reproduce the time-series data (**b (ii)**), the values of α and β are similar and positive (**b (ii)**) indicating that the regulation from A to B is a mixture. Thus, the model-based method fails to infer the network structure. **c** We then apply GOBI to infer the network structure. To apply GOBI, we first use the moving-window technique with a window size of 10 and an overlapping ratio of 10% to divide the time-series data of A and B into 41 different segments. Then, from each time-series data, $S_{X\sigma}^Y$ is computed for every 1D and 2D regulation (**c (i)-(ii) left**). The criteria $S_{X\sigma}^Y = 1$ infers no 1D regulation (**c (i) right**), and one 2D regulation (**c (ii) right**). This 2D regulation passes the Δ test and GOBI successfully infers the network structure.

6. Are the monotonic models as investigated in the current work sufficient representative? Are the non- or partially-monotonic models are more prevalent? These justifications also should be further provided. Is there any chance to extend the current work to those non-monotonic or even temporal-structured models?

First, in this study, we have investigated various examples of monotonic systems at different levels: molecular levels (genetic oscillator, Fig. 4b; repressilator, Fig. 4c; and cofactors at estradiol sensitive promoter, Fig. 4d), organism level (prey-predator system, Fig. 4a), and climate level (air pollutants and cardiovascular disease, Fig. 4e). In particular, these examples were used to test previous causation detection algorithms (Sugihara, G. et al., Science, 2012; Leng, S. et al., Nat Commun, 2020; Tyler, J. et al., Bioinformatics, 2022; Pigolotti, S. et al., Proc. Natl. Acad. Sci. USA, 2007). Thus, we believe that the examples used in our current work well represent monotonic systems in the real world.

Determining whether monotonic systems are more prevalent than non- or partially-monotonic systems is challenging. However, it is evident that monotonic regulation plays a crucial role at various scales. First, at the molecular level, gene regulation is often monotonic. For example, the binding (unbinding) of specific proteins, known as activators (repressors), to DNA, is necessary for transcription to begin (stop), indicating positive (negative) causal interaction. Additionally, there are numerous monotonic relationships among molecules, such as hormones (Waadt, R. et al., Nat Rev Mol Cell Biol, 2022; Mazziotti, G., Giustina, A., Nat Rev Endocrinol, 2013) and metabolites (Baker, S.A., Rutter, J., Nat Rev Mol Cell Biol, 2023). Next, in epidemiology, certain variables can positively (or negatively) influence related diseases. For example, standing water serves as a breeding ground for mosquitoes that transmit the dengue virus, so rainfall positively affects the incidence of dengue fever. Also, high temperatures expedite mosquito development, leading to a positive relationship between temperature and the outbreak of dengue fever (Alto BW, Bettinardi D., The American Society of Tropical Medicine and Hygiene, 2013; Lai, YH., BioMed Eng OnLine, 2018). Lastly, monotonic relationships are

observed in the climate system, such as a strong positive feedback effect between temperature and greenhouse-gas concentrations (Egbert H. et al., Nat. Clim. Change, 2015).

Nevertheless, we agree with the reviewer that non- or partially-monotonic systems also play important roles in the real world. Thus, we have now illustrated how GOBI can be potentially used to detect non-monotonic regulation. Specifically, we have described how GOBI can identify the types of regulation in a temporal-structured system, where the regulation type is changed from positive regulation to non-monotonic regulation, negative regulation and the absence of regulation, as described below (Supplementary Fig. 11). We have also discussed this issue in the Discussion section (see Reviewer 3's comment 2).

- Supplementary Information:

Section XIV. Extension to temporal-structured model including non-monotonic regulation

Here, we illustrate an example of how GOBI can be applied to a temporal-structured system that includes non-monotonic regulation. To construct a temporal-structured regulation from A to C (Supplementary Fig. 11a (i)), the values of parameters (α , β , and γ) are varied over time (Supplementary Fig. 11a (ii)), resulting in different types of regulation from A to C: positive regulation in the first quarter, non-monotonic regulation in the second quarter, negative regulation in the third quarter, and the absence of regulation in the last quarter. We simulate 100 time-series data of C from different input signals A and B (Supplementary Fig. 11b). The time series of input signals are constructed by randomly selecting 200 points from [0, 1] over the time domain and connecting them with the 'spline' method.

Next, we use a moving-window technique with a window size of 5 and an overlapping ratio of 50% to segment the time-series data of A, B, and C. For each window, regulation-detection scores for positive and negative regulation ($S_{A^+C^-}^C$ and $S_{A^-C^-}^C$) are computed from the 100-time series (Supplementary Fig. 11c). The criterion $S_{A^+C^-}^C = 1$ ($S_{A^-C^-}^C = 1$) is satisfied in the presence of positive (negative) regulation (Supplementary Fig. 11c, the first and third quarter). However, in the presence of non-monotonic regulation or the absence of regulation, neither criterion is satisfied, indicating that the regulation-detection score cannot distinguish between these two cases (Supplementary Fig. 11c, the second and last quarter). In particular, when the strength of both positive and negative regulation is similar, the regulation-detection scores are around zero (the middle of the second quarter in Supplementary Fig. 11c), which is similar to the case of the absence of regulation.

On the other hand, the regulation-detection functions, instead of their normalized integrals (i.e., regulation-detection scores) differ between non-monotonic regulation and the absence of regulation. To distinguish between these cases, we utilize the regulation-detection functions $I_{A^+C^-}^C$ and $I_{A^-C^-}^C$, which correspond to positive and negative regulation from A to C, respectively. These functions, $I_{A^+C^-}^C$ and $I_{A^-C^-}^C$, are defined on the regulation-detection regions $R_{A^+C^-}$ (bottom-right triangle, Supplementary Fig. 11d) and $R_{A^-C^-}$ (top-left triangle, Supplementary Fig. 11d),

respectively. We compute $I_{A^+C^-}^C$ and $I_{A^-C^-}^C$ for each time series window using all available data. As expected, the presence of positive and negative regulations resulted in positive values for $I_{A^+C^-}^C$ and $I_{A^-C^-}^C$ on whole their domains, respectively (Supplementary Fig. 11e (i) and (iii)). On the other hand, in the presence of non-monotonic regulation or the absence of regulation, the regulation-detection functions have both positive and negative values (Supplementary Fig. 11e (ii) and (iv)). However, the patterns of the mixture of positive and negative values are completely different between non-monotonic regulation (Supplementary Fig. 11e (ii)) and the absence of regulation (Supplementary Fig. 11e (iv)). Specifically, in the absence of regulation, the sign of the regulation-detection function varies inconsistently across the time-series data, resulting in a completely mixed sign of the regulation-detection function (Supplementary Fig. 11e (iv)). In contrast, in the presence of non-monotonic regulation, the sign of the regulation-detection function is consistent across the time-series data (Supplementary Fig. 11e (ii)). Taken together, $I_{A^+C^-}^C$ and $I_{A^-C^-}^C$ are consistently positive or negative at the specific values of $(A(t), A(t^*))$ in the presence of non-monotonic regulation, but not in the absence of regulation.

We quantify this consistency to distinguish between non-monotonic regulation and the absence of regulation. We first divide the regulation-detection regions ($R_{A^+C^-}$ and $R_{A^-C^-}$) into triangles of equal shape (Supplementary Fig. 11f left). Then, we count the number of positive values of $I_{A^+C^-}^C$ and $I_{A^-C^-}^C$ at each triangle, and normalized this count by the total number of positive values to approximate the probability distribution of positive values (Supplementary Fig. 11f right, P). Similarly, we approximate the probability distribution of negative values (Supplementary Fig. 11f right, N). Here, we use the results of regulation-detection functions on the window in the presence of non-monotonic regulation as an example (Supplementary Fig. 11e (ii)). Finally, we quantify the similarity between the probability distribution of positive and negative values (P and N) using the Jensen-Shannon Divergence (JSD) (Supplementary Fig. 11f bottom-right):

$$JSD(P \parallel N) = \frac{1}{2}KL(P \parallel Q) + \frac{1}{2}KL(N \parallel Q), \text{ where } Q := \frac{1}{2}(P + N).$$

A high (low) value of JSD indicates that the regulation-detection function has consistent (inconsistent) sign throughout the region. The similarity, measured by JSD, is much higher when non-monotonic regulation is present compared to the absence of regulation (Supplementary Fig. 11g). This suggests that the consistency test based on JSD can be used to distinguish between non-monotonic regulation and the absence of regulation.

We next investigate whether the proposed consistency test can distinguish between non-monotonic regulation and the absence of regulation when experimental data is given. Specifically, among the eight different time-series data of the genetic oscillator (Fig. 4b), we use four time series with a similar range of σ^{28} (Supplementary Fig. 11h). Using these data, we test the consistency of the signs of regulation-detection functions for three types of regulations: positive regulation of σ^{28} for TetR, non-monotonic regulation of σ^{28} for $e^{TetR} \times \sigma^{28}$, and the absence of regulation. The regulation of σ^{28} for $e^{TetR} \times \sigma^{28}$ is non-monotonic due to the positive regulation from σ^{28} to TetR and negative self-regulation of σ^{28} . Furthermore, we investigate the

regulation of σ^{28} for TetR* for the absence of regulation, where TetR* is measured under different conditions than σ^{28} . For the positive regulation, $I_{\sigma^{28}^+ \text{TetR}^-}^{\text{TetR}}$ is always positive (Supplementary Fig. 11i ①, bottom-right triangle). For the non-monotonic regulation, $I_{\sigma^{28}^- \text{TetR}^-}^{\text{TetR} \times \sigma^{28}}$ and $I_{\sigma^{28}^+ \text{TetR}^-}^{\text{TetR} \times \sigma^{28}}$ have both positive and negative values, while their signs are consistent across the time-series data (Supplementary Fig. 11i ②). On the other hand, in the absence of regulation, the sign of the regulation-detection function varies across the data (Supplementary Fig. 11i ③). Consequently, JSD is much higher for non-monotonic regulation than for the absence of regulation (Supplementary Fig. 11j ② and ③), indicating that the consistency test can distinguish between non-monotonic regulation and the absence of regulation.

Supplementary Fig. 11. Extended framework of GOBI to distinguish non-monotonic regulation from the absence of regulation. **a** As described by the ODE (a (i)), the time-varying values of parameters (α , β , and γ) (a (ii)) result in different types of regulation from A to C: positive regulation in the first quarter, non-monotonic regulation in the second quarter, negative regulation in the third quarter, and the absence of regulation in the last quarter. **b** With the ODE describing the network (a (i)), 100-time series are simulated from different initial conditions and one of them is shown. **c** Regulation-detection scores are computed for positive and negative

regulations ($S_{A^+C^-}^C$ and $S_{A^-C^-}^C$), and the mean value and standard deviation over 100 –time-series data are shown. The criterion $S_{A^+C^-}^C = 1$ ($S_{A^-C^-}^C = 1$) is satisfied in the presence of positive (negative) regulation, whereas neither is satisfied in the presence of non-monotonic regulation or the absence of regulation. **d-e** Non-monotonic regulation and the absence of regulation can be distinguished by comparing regulation-detection functions ($I_{A^+C^-}^C$ and $I_{A^-C^-}^C$). **d** $I_{A^+C^-}^C$ and $I_{A^-C^-}^C$ are defined on the regulation-detection regions $R_{A^+C^-}$ (bottom-right triangle) and $R_{A^-C^-}$ (top-left triangle), respectively. **e** The regulation-detection functions, computed from 100 –time-series data for each regulation-detection region, are shown with the color indicating its sign. $I_{A^+C^-}^C$ and $I_{A^-C^-}^C$ are always positive in the presence of positive and negative regulation (**e** (i) bottom-right triangle and (iii) top-left triangle), respectively. On the other hand, in the presence of non-monotonic regulation and the absence of regulation, $I_{A^+C^-}^C$ and $I_{A^-C^-}^C$ have both positive and negative values (**e** (ii) and (iv)). While the regulation-detection functions are always positive or negative at the specific region across all the time-series data for non-monotonic regulation (**e** (ii)), the region where regulation-detection functions are positive or negative keeps changing across the data for the absence of regulation (**e** (iv)). **f** To quantify this consistency, regulation-detection regions are triangularized (**f** left). Next, the approximated probability distribution of the positive values (**f** right, P) and the negative values (**f** right, N) of $I_{A^+C^-}^C$ and $I_{A^-C^-}^C$ are obtained. Then, the Jensen-Shannon Divergence (JSD) is used to measure the similarity between two probability distributions (**f** bottom-right). Here, 32 number of partitions are used. Also, to illustrate the P and N , the results of regulation-detection functions on the window in the presence of non-monotonic regulation are used (**e** (ii)). **g** The JSD is higher in the presence of non-monotonic regulation than in the absence of regulation. **h-j** This strategy is applied to the time-series data of the genetic oscillator (**h**) by comparing the regulation-detection function in the presence of positive regulation (**i** ①), non-monotonic regulation (**i** ②), and the absence of regulation (**i** ③). JSD is higher in the presence of non-monotonic regulation than in the absence of regulation (**j**).

Reviewer: 2

The authors attempted to devise a general ODE framework to infer causal relationships between variables from time-series data, with the aim to circumvent limitations of model-free methods on false discoveries and to expedite computation over existing model-based methods that fit various families of specific functions to data so that the underlying mechanistic relationship can be learned. Pivoting on the assumption of monotonic pairwise relationship, the method developed here (GOBI) was shown to detect directed regulation and rule out indirect regulation. The authors tested the efficacy of their method on simulated datasets as well as experimental datasets, claiming that GOBI is accurate and broadly applicable.

Recent years has witnessed a number of endeavors for developing a general ODE framework to infer causal relationship from time-series data (PMID: 27698038, 34675223, 36476735). The theoretical development presented in this study is straightforward but refreshing. If done properly, it could very well be a generic tool for causal inference using time-series data. Unfortunately, the current manuscript has several fundamental issues, raising concerns about the validity and applicability of the proposed method. Below are some general comments.

We thank the reviewer the positive feedback about the potential of GOBI as a generic tool for causal inference. Furthermore, we appreciate the reviewer for constructive and detailed comments, which greatly improves the applicability and readability of the manuscript. We have also cited the suggested relevant works in the manuscript as follows:

Section I: "... However, the inference results strongly depend on the choice of model, and inaccurate model imposition can result in false positive predictions, limiting their applicability. To overcome this limit, inference methods using flexible models were developed [30-36; Xie, X., Samaei, A., Guo, J. et al., *Nat Commun*, 2022; Chen, Z., Liu, Y. & Sun, H., *Nat Commun*, 2021; and Tegnér, J. et al., *Philosophical Transactions of the Royal Society A: Mathematical, Physical and Engineering Sciences*, 2016]. ..."

1. The assumption of monotonicity greatly limits the applicability of GOBI. It is ubiquitous that biological variables interact to influence their dynamics, meaning that monotonic relationship (Equation 1) is extremely rare in real-world scenarios. Since this assumption serves as the foundation for causal inference in GOBI (see Supporting Information Section 1), violations could also confine the validity of the method. Consider a predator-prey model with an interaction term ($Y' = X*Y - Y$). Due to the non-monotonicity in X and Y' and the non-independence between X and Y , theorem 1 does not hold and the regulation detection score is bound below 1. This ensures false negative discoveries. Moreover, multivariable monotonicity has not been defined formally.

We acknowledge the reviewer's concerns about the assumption of monotonicity in GOBI, which could potentially limit its applicability. However, it is important to note that the primary goal of GOBI is not only to infer causality, but also to determine the type of causation, whether positive

or negative. To achieve this, we have focused on monotonic systems. Moreover, monotonic relationships are common across various scales (see Reviewer 1's comment 6 for details).

Nevertheless, we have now extended GOBI to distinguish between non-monotonic regulation and the absence of regulation (see Reviewer 1's comment 6 for details), which can increase the applicability of our method in cases where the assumption of monotonicity is not valid. For example, in the case of the prey-predator model, the predator population (Y) dynamics can be described as $\dot{Y} = f(X, Y) = X \cdot Y - Y$, where X represents the prey population. As X and Y are both positive, f is a monotonic increasing function with respect to X and a non-monotonic function with respect to Y , reflecting the growth and death of the population. Using our current algorithm, we can only identify the monotonic regulation from X to Y . However, with our extended algorithm, we can also detect the non-monotonic regulation of Y itself, which is crucial for capturing the complex dynamics of the system.

Also, as the reviewer pointed out, multivariable monotonicity has not been defined formally. We unintentionally omitted a clear description of multivariable monotonicity. To resolve this, we have now revised the manuscript as follows:

- Supplementary Information (Section I):

"If there is a ND regulation from $X = (X_1, X_2, \dots, X_N)$ to Y , then the dynamic of Y is given as

$$\frac{dY}{dt} = f(X) = f(X_1, X_2, \dots, X_N).$$

For each $i \in \{1, 2, \dots, N\}$, we assume that X_i either positively ($X_i \rightarrow Y$) or negatively ($X_i \dashv Y$) regulates Y . If X_i positively (negatively) regulates Y , f is monotonically increasing (monotonically decreasing) with respect to X_i . For example, if X_i positively regulates Y , then for all fixed $\tilde{x}_j \in \text{range}(X_j)$ where $j = 1, 2, \dots, N$ except for i , $f_i(\tilde{X}_j) : x_i \in \text{range}(X_i) \mapsto f(\tilde{x}_1, \dots, \tilde{x}_{i-1}, x_i, \tilde{x}_{i+1}, \dots, \tilde{x}_N)$ is monotonically increasing. Conversely, if X_i negatively regulates Y , $f_i(\tilde{X}_j)$ is monotonically decreasing."

2. Most examples illustrated in the manuscript have the form of linear combination of single-variable monotonic functions (Fig. 1, Fig. 2a/b/d/e, and Fig. 3). It seems the authors were trying to avoid the aforementioned issues. For the cAMP oscillator (Fig. 2f), where there were interaction terms, the supplementary data indicated that GOBI detected a false regulation ($ACA \rightarrow \text{cAMPe}$) and missed a strong prediction ($ACA \rightarrow \text{cAMPi}$). These mishaps were omitted in the main figure and not discussed in the main text. Nevertheless, the authors claimed that they recovered the relationship network successfully, prompting necessary doubt over their implementation. Further, the part of inference with noisy data (Fig. 3) was not included in the code, making it impossible to verify or replicate.

We acknowledge that most of our examples in the manuscript contained only a linear combination of single-variable monotonic functions. However, we did not intend to avoid the inevitable issues associated with multivariable monotonicity, but have merely chosen the popular biological oscillatory models, which were considered in previous inference studies (Tyler, J., Forger, D. & Kim, J. K., *Bioinformatics*, 2022; Pigolotti, S., Krishna, S., & Jensen, M. H., *Physical review letters*, 2009). In the presence of interaction terms (i.e., multivariable monotonicity), GOBI can still infer a regulatory network. For instance, let us consider a network including three different interaction terms (Fig. R1a): the multiplication of two variables ($A \cdot C^3$), the multiplication of a hill function and a single variable ($\frac{1}{0.01+B^4} \cdot D$), and the multiplication of two different hill functions ($\frac{A^4}{1+A^4} \cdot \frac{1}{0.01+C^4}$). With the ODE describing the network, we simulate 100 time-series data from different initial conditions. Next, from each time series, we compute the regulation-detection score for every 1-3D regulation (Fig. R1b (i)-(iii) left). The criteria $S_{X\sigma}^Y = 1$ infers no 1D regulation (Fig. R1b (i) right), three 2D regulations (Fig. R1b (ii) right), and six 3D regulations (Fig. R1b (iii)). However, all inferred 3D regulations are identified as false positive predictions via the Δ test because $\Delta_{A+C}^B(D) = 0$, $\Delta_{B+D}^C(A) = 0$, and $\Delta_{A+C}^D(B) = 0$. By merging the inferred 2D regulations, the original regulatory network can be successfully inferred (Fig. R1a (ii)) even in the presence of interaction terms.

Figure R1. Inferring a regulatory network in the presence of interaction terms. **a** The ODE (a (i)) describes a regulatory network (a (ii)) with three different interaction terms: the multiplication of two variables ($A \cdot C^3$), the multiplication of a hill function and a single variable ($\frac{1}{0.01+B^4} \cdot D$), and the multiplication of two different hill functions ($\frac{A^4}{1+A^4} \cdot \frac{1}{0.01+C^4}$). With the ODE describing the network, various time series are simulated from different initial conditions. The time series of input signal A is constructed by connecting 21 randomly selected points from $[0, 1]$ over the time domain using spline fitting. Also, the initial conditions for B , C , and D are randomly selected from $[0, 1]$. **b** From each time series, $S_{X\sigma}^Y$ is computed for every 1-3D regulation (b (i)-(iii) left). The criteria $S_{X\sigma}^Y = 1$ infers no 1D regulation (b (i) right), three 2D regulations (b (ii) right), and six 3D regulations (b (iii) right). However, all inferred 3D regulations are identified as false-positive

predictions via the Δ test. For example, $A \rightarrow C \rightarrow D \rightarrow B$ and $A \rightarrow C \rightarrow D \nrightarrow B$ are false-positive because $\Delta_{A^+C^+}^B(D) = 0$. By merging the inferred 2D regulations, GOBI successfully infers the network structure in the presence of interaction terms.

Next, the reviewer pointed out that our supplementary data indicates that GOBI detected a false regulation ($ACA \rightarrow cAMP_e$) and missed a strong prediction ($ACA \rightarrow cAMP_i$). We believe the reviewer may have misinterpreted our results. First, $ACA \rightarrow cAMP_e$, which was inferred by GOBI, is not a false regulation [Maeda, M. et al., Science, 2004]. Second, we believe the reviewer thought that GOBI did not infer $ACA \rightarrow cAMP_i$ from 1D inference results, but it was inferred via 2D inference (i.e., $ACA \rightarrow RegA \nrightarrow cAMP_i$). Specifically, to understand the Supplementary Data 1, it is important to note that GOBI infers multi-dimensional regulations, which implies that each ND framework of GOBI only infers N dimensional regulations. As the reviewer mentioned, $ACA \rightarrow cAMP_i$ was not inferred based on the results of the 1D framework in Supplementary Data 1. However, when the 2D framework was performed, $ACA \rightarrow cAMP_i$ was inferred as a component of the 2D regulation $ACA \rightarrow RegA \nrightarrow cAMP_i$. Thus, $ACA \rightarrow cAMP_i$ was included in the inferred network structure.

Nevertheless, while reporting the results, we unintentionally presented the positive regulation from ACA to $cAMP_e$ as negative in Figure 2f. Thus, we have revised the Figure 2f as follows.

Lastly, in response to the reviewer's concern regarding the code for Figure 3, we have now submitted all the codes used in our manuscript with detailed annotations (see Reviewer 2's comment 4 for details).

3. The manuscript also contains multiple cases of inconsistency and contrived arguments, severely reducing readers' confidence in the contents. For instance, on page 5, the authors mentioned that noise made it difficult for the regulation-detection score to distinguish direct and indirect regulations, thus they needed another criterion for inference, i.e., the surrogate test and combining the p-values. Occasions like this leave one to ponder how limited the method actually is.

Our framework, GOBI, comprises three main steps: regulation-detection score, delta test, and surrogate test. In order to enhance readability, we have explained our framework in a step-by-step manner throughout the current manuscript. In Fig. 1 and 2, we demonstrated that the regulation-detection score and delta test are sufficient to infer regulations from simulated time-series data in the absence of noise. However, in the presence of noise, we introduced the

surrogate test as an additional step to handle noisy data. Thus, the sentence mentioned by the reviewer does not indicate a limitation of GOBI, but rather serves as a transition to explain the incorporation of the surrogate test into GOBI.

4. It is far from clear how GOBI would be a user-friendly tool for the wider community. The analyses on simulated datasets (Fig. 2) appear to involve ad hoc selection that requires domain-specific knowledge or personal discretion (see supplementary data). Inference using experimental data (Fig. 4) was not described in details, and there was no associated supplementary data. It is not entirely obvious how the authors arrived at their findings, other than the schematic summaries. In addition, the code was not annotated for easy adoption and the user's manual is missing.

As the reviewer mentioned, the analysis on simulated data sets involved assuming negative self-regulation as prior knowledge. However, assuming the types of self-regulations is just a useful option for users to accurately infer network structures with limited data. GOBI can successfully infer the network structures without assuming negative self-regulation. We have now illustrated this more clearly (see Reviewer 3's comment 4). Additionally, we have provided detailed explanations of the inference process using experimental data in the Supplementary Information, including results from three different steps in GOBI: total regulation score, delta test, and surrogate test (Supplementary Fig. 9).

Nevertheless, we agree with the reviewer's comment that our codes were not annotated for easy adoption. To address this concern, we have now submitted all the codes used in our manuscript with detailed annotations. Furthermore, we have now included a user manual with a simple example to help users adopt GOBI more easily. Now, to use GOBI, users need to do the following tasks after uploading the data, which are easy to do, as seen below.

1) Users can optionally choose the thresholds for the regulation-detection region, regulation-detection score, and total regulation score, as well as the critical values for the Δ test and the surrogate test. To assist users in selecting those values, we have provided guidelines (see Reviewer 2's comment 7 and Reviewer 3's comment 6 for details). Thus, users can use our guidelines as defaults for inference or make adjustments depending on whether the goal is to decrease false positive or negative predictions.

2) Users can optionally incorporate the available prior knowledge into the inference (see Reviewer 3's comment 4 for details). Of course, it is possible to run the inference without any prior knowledge, but incorporating such knowledge is beneficial.

3) Users can select the maximum dimension of the framework for inference. Typically, if the system of interest consists of N components, it is recommended to run the inference up to an $N - 1$ dimensional framework (N dimensional framework including self-regulations). However, this recommendation is not always feasible because inferring high-dimensional regulation requires a large amount of data. To assist users in choosing an appropriate dimension of the framework, we provided guidelines to check whether the current data is sufficient for running the

inference of a specific dimension (see Reviewer 3's comment 10). Thus, users can easily select the maximum possible dimension of the framework based on the available data.

Based on this, we have now included the user manual in the Supplementary Information as follows:

- Supplementary Information:

Section XVIII. Manual for the GOBI computational package

Here, we provide a manual for a computational package, GOBI (General ODE-Based Inference), to infer a network structure from time-series data. To help user's understanding, we use an illustrated example of the repressilator (Fig. 4c).

1. Generate the 'data.mat' file, which contains two variables 't' and 'y'. 't' is the time points at which the measurements were taken. Each column of 'y' should be the data for each variable at the respective time points (see Input in Supplementary Fig. 14).

2. Run the 'Step0_interpolation_and_cut.m' file (see Interpolation & Cut in Supplementary Fig. 14). This function interpolates the data based on the interpolation method specified by the user and cuts the data into windows using moving window technique, where the window size and overlapping ratio are defined by the users as follows.

(a) Users need to specify the interpolation method using the parameter 'method'. Specifically, 'method = 1' indicates 'linear' interpolation, 'method = 2' indicates 'spline' interpolation, and 'method = 3' indicates 'fourier' interpolation. In the case of 'method = 3', users also have to specify the order of 'fourier' interpolation (i.e., 'num_fourier = 1 to 8'). For less noisy data, 'spline' method is recommended, and for highly noisy data, 'fourier' method with order 2 is recommended.

(b) Users need to choose the sampling rate for the interpolation. The parameter 'time_interval' indicates how finely the users wants to interpolate the original time series. For example, 'time_interval = 0.5' indicates interpolation using a time interval twice as fine as the original time series. Selecting 'time_interval' to make about 100 time points per period is recommended, and please note that the low value of 'time_interval' (high sampling rate) makes the inference accurate, but slow as well.

(c) For the data segmentation, users need to specify the parameters for the moving window technique, i.e., window size and overlapping ratio. The parameter 'window_size_ori' defines the number of time points in each window. Then, along the time series, we move the window until the next window overlaps with the current window by the ratio defined in the parameter 'overlapping_ratio' ('overlapping_ratio = 0.1' as a default). For oscillatory time-series data, it is recommended to choose the window size as one period. The time series in every window is saved at the variable 'y_total'.

After interpolation and data segmentation, the data is saved in 'data_cut.mat' file.

3. Update the 'Step0_options.m' file (see Options in Supplementary Fig. 14). This code integrates options that can be adjusted by the users or set via our guidelines.

(a) Users need to specify the thresholds for regulation-detection region ('thres_R'), regulation-detection score ('thres_S'), and total regulation score ('thres_TRS') as well as the critical values for Δ test ('p_delta = 0.01' as defaults) and surrogate test ('p_surrogate = 0.001' as defaults). To assist users in selecting those values, we have provided guidelines based on the noise level of data (Supplementary Fig. 5). Thus, users can use our guidelines as defaults or make adjustments depending on whether the goal is to decrease false positive or negative predictions.

(b) The parameter 'type_self' defines options for the types of self-regulation: no assumption ('type_self = NaN'); negative self-regulation ('type_self = -1'); no self-regulation ('type_self = 0'); and positive self-regulation ('type_self = 1'). Also, users can optionally incorporate other available prior knowledge into the inference (Supplementary Fig. 3). Of course, it is possible to run the inference without any prior knowledge, but incorporating such knowledge is beneficial when the amount of data is limited.

(c) The parameter 'max_D' defines the maximum dimension of the framework for inference. Typically, if the system of interest consists of N components, it is recommended to run the inference up to an $N - 1$ dimensional framework (N dimensional framework including self-regulations). However, this recommendation is not always feasible because inferring high-dimensional regulation requires a large amount of data. To assist the users in choosing an appropriate dimension of the framework, we have provided the guidelines to check whether the current data is sufficient for running the inference of a specific dimension (Supplementary Fig. 13).

All the options are saved in the 'data_with_options.mat' file. Also, during the inference, our framework automatically gives a warning signal when the data is insufficient to run the framework. Then, users should adjust these options.

4. Run the codes for 1D framework (see Step 1 in Supplementary Fig. 14).

(a) Run the 'Step1_compute_RDS_dim1.m' function. First, this function finds all the possible 1D regulations and saves them at the variable 'component_list_dim1'. Each row of 'component_list_dim1' indicates the set of causal variable (C) and target variable (T). For each pair (C and T), regulation-detection region and score are computed for all the regulation types (+ and -) using time-series data, and they are saved at the variables 'R_total_list' and 'S_total_list'. Those values are saved in the 'RDS_dim1.mat' file.

(b) Run the 'Step1_compute_TRS_dim1.m' function. Using the 'thres_R' and 'thres_S' that users specified, Total Regulation Score (TRS) is computed for each possible 1D regulation. As a result, the heatmap of TRS is displayed, and the exact values of TRS are saved at the variable 'TRS_total' in the 'TRS_dim1.mat' file. In this heatmap, each row indicates the possible 1D regulation (C and T) and each column indicates the regulation type (+ and -). Using the 'thres_TRS' that users specified, 1D regulations are inferred.

(c) Run the 'Check1D.m' function. This function checks whether the data is sufficient to confidently infer 1D regulations. If the warning signal comes out, users are recommended to stop the inference or adjust the options.

5. Run the codes for 2D framework (see Step 2 in Supplementary Fig. 14).

(a) Run the 'Step2_compute_RDS_dim2.m' function. First, this function finds all the possible 2D regulations and saves them at the variable 'component_list_dim2'. Each row of 'component_list_dim2' indicates the set of two causal variables (C_1 and C_2), and target variable (T). For each set (C_1, C_2, T), regulation-detection region and score are computed using time-series data for all the regulation types ((+, +), (+, -), (-, +), and (-, -)) and they are saved at the variables 'R_total_list' and 'S_total_list'. These values are saved in the 'RDS_dim2.mat' file.

(b) Run the 'Step2_compute_TRS_dim2.m' function. Using the 'thres_R' and 'thres_S' that users specified, Total Regulation Score (TRS) is computed for each possible 2D regulation. As a result, the heatmap of TRS is displayed, and the exact values of TRS are saved at the variable 'TRS_total' in the 'TRS_dim2.mat' file. In this heatmap, each row indicates the possible 2D regulation (C_1, C_2, T) and each column indicates the regulation type ((+, +), (+, -), (-, +), and (-, -)). Using the 'thres_TRS' that users specified, candidates for 2D regulations are inferred.

(c) Run the 'Step2_Delta_test_dim2.m' function. For every candidate for 2D regulations (Inferred from 5-(b)), this function performs the Δ test for each causal variable (C_1 and C_2). If the number of data is smaller than 25, then this function tests whether the signs of regulation-delta functions are non-negative or not. If the number of data is larger than 25, this function performs the Wilcoxon signed ranked test. The result of Δ test is saved at the variable 'delta_list' in the 'Delta_dim2.mat' file. Each row of the 'delta_list' represents the candidate for 2D regulation, and two columns of 'delta_list' represents the results of Δ test for C_1 and C_2 . Using the 'p_delta' that users specified, candidates for 2D regulations are inferred.

(d) Run the 'Step2_Surrogate_test_dim2.m' function. For every candidate for 2D regulations (inferred from 5-(c)), this function performs the surrogate test for each causal variable. Users need to specify the number of bootstrapping ('num_boot = 100' as default) for the surrogate test. During the simulation, for every time-series data, one of causal variables (C_1 or C_2) is shuffled 'num_boot' times and regulation-detection scores are computed. Then the p-value is computed for each data and causal variable. Those p-values are combined using Fisher's method. The results of the surrogate test are saved in the variable 'surrogate_list' in the

'Surrogate_dim2.mat' file. Each row of 'surrogate_list' represents the candidate for 2D regulations. The first two columns of 'surrogate_list' represent the results of surrogate test for C_1 and C_2 . The third and fourth columns of 'surrogate_list' represent the thresholds for combined p-values (combine 'p_surrogate' for all the data). Finally, by using these thresholds, 2D regulations are inferred.

(e) Run the 'Check2D.m' function. This function checks whether the data is sufficient to confidently infer 2D regulations. If the warning signal comes out, users are recommended to stop the inference or adjust the options.

These steps are continued until the 'max_D'-dimensional framework. After that, run the function 'Merging_regulations.m' to infer a network structure by merging all the inferred regulations. Since GOBI involves multi-dimensional inferences, it is possible to detect various dimensional regulations for a single target. In this case, GOBI infers the regulation with the highest value of TRS. Here, we illustrate up to the 2D framework, but users can easily expand this approach to include higher dimensions as needed (see Github codes (<https://github.com/Mathbiomed/GOBI>) for Fig. 4).

Supplementary Fig 14. Sample input and output for the GOBI package based on the experimental repressilator example (Fig. 4b). The 'data.mat' file contains the time points ('t') and the time-series data for each variable ('y'). Then, the time series are interpolated using the 'spline' method and cut using the moving window technique. Next, users have options regarding hyper-parameters, types of self-regulation, and the maximum dimension of the framework. Here, we use default values for hyper-parameter, assuming negative self-regulation, and specifying 'max_D = 2'. With these options, our 1D framework computes TRS and infers 1D regulations. Also, our 2D framework computes TRS and performs Δ test and surrogate test to infer 2D regulations. By merging the inferred regulations, a network structure is reconstructed. During the inference, our framework automatically gives a warning signal when the data is insufficient.

Specific Comments:

5. Fig. 1a: Notations are misplaced. Z^d should be X^d , and X^d should be Y^d
6. Fig.1d-l: It is not clear what the color bar refers to.

We thank the reviewer for bringing to our attention the misplaced and omitted notations in Figure 1. The color bars in the figure represent the value of the regulation-detection functions. Based on this, we have revised Figure 1 as follows:

7. Fig. 2a and 3i: Why was the critical value for the Delta test 0.01 and 0.001 for the surrogate test? Were they derived from benchmarking?

The critical values for the Delta test and surrogate test were not derived from the benchmarking study. Instead, we determined the optimal critical values for inferring true network structures from noisy simulated data (Fig. 3). Then, we used these optimized values to infer causation from experimental time-series data (Fig. 4 and 5). Note that the critical value for the surrogate test is more stringent than that of the Delta test because the surrogate test aims to distinguish between direct and indirect regulations, which the Delta test cannot do in the presence of high levels of noise (Supplementary Fig. 6). While we recommend using these values as the default in GOBI, depending on whether the goal is to decrease false positive or negative predictions, one can adjust the threshold (i.e., choose more or less strict critical values).

8. Fig. 3c: The blue line should be 1 for low noise levels according to Fig. 3b. It is not addressed why the regulation of $A \rightarrow C$ was not detected using TRS thres.

We thank the reviewer for pointing out the errors in Figure 3. As the reviewer mentioned, the blue line, representing the total regulation score for $A \rightarrow B$, should be one for low noise levels. Based on this, we have revised Figure 3 as follows.

Next, as pointed out by the reviewer, the positive regulation $A \rightarrow C$ did not meet the 1D criteria for TRS ($TRS_{X\sigma}^Y > TRS^{thres}$). Thus, our 1D framework only inferred $A \rightarrow B$, which met the criteria for TRS. However, after applying the 2D framework, 2D regulation $A \rightarrow B \rightarrow C$ is inferred, which met the 2D criteria for TRS. By merging the inferred 1D and 2D regulations, we can successfully reconstruct the network structure of IFL. This highlights the need for multi-dimensional inferences as $A \rightarrow C$ can be inferred with 2D inference but not 1D inference. To help the reader's understanding, we have now revised the manuscript as follows:

Section II. B: "... By merging the inferred 1D and 2D regulations, the regulatory network is successfully inferred. Here, note that the regulation $A \rightarrow C$ is not detected by the 1D regulation-detection score since C has multiple causes. However, the 2D regulation-detection score detects $A \rightarrow B \rightarrow C$, which contains $A \rightarrow C$. This demonstrates the need for multi-dimensional inferences, as the 1D criteria alone would not have been sufficient to fully capture the regulatory relationships in the network. Since this system has three components, we infer up to 2D regulations. If there are N components in the system, we go up to $(N - 1)$ D regulations (Supplementary Fig. 2)."

Section II. C: "In the presence of noise in the time-series data, the regulation-detection score ($S_{X\sigma}^Y$) is perturbed. Thus, $S_{X\sigma}^Y$ may not be one even if there is a regulation type σ from X to Y . For example, in the case of an Incoherent Feed-forward Loop (IFL) which contains $A \rightarrow B$ (Fig. 3a), S_A^B is always one in the absence of noise (Fig. 2a Step 1 blue), but not in the presence of noise (Fig. 3b blue). Thus, for noisy data, we need to relax the criteria $S_{X\sigma}^Y = 1$ to $S_{X\sigma}^Y > S^{thres}$ where

$S^{thres} < 1$ is a threshold. Because S_{A-}^B gets farther away from one as the noise level increases, S^{thres} should also be decreased accordingly. We choose S^{thres} as $0.9 - 0.005 \times (\text{noise level})$ with which true and false regulations can be distinguished in the majority of cases for our previous *in silico* examples (Fig. 3b and Supplementary Fig. 5e). For instance, S^{thres} (green dashed line, Fig. 3b) overall separates true regulation (Fig. 3b blue) and false regulation (Fig. 3b red). Here, we choose $A \rightarrow C$, which has the highest score among all false positive 1D regulations (Fig. 2b red).

We found that the fraction of time-series data satisfying $S_{X\sigma}^Y > S^{thres}$, which we refer to as the Total Regulation Score (TRS) (Fig. 3c left), more clearly distinguishes the true (Fig. 3c right, blue) and false positive (Fig. 3c right, red) regulations. Thus, we use the criteria $TRS_{X\sigma}^Y > TRS^{thres}$ to infer the regulation. Similar to S^{thres} , TRS^{thres} also decreases as the noise level increases. Specifically, we use $TRS^{thres} = 0.9 - 0.01 \times (\text{noise level})$, which successfully distinguishes between the true and false regulations of IFL (Fig. 3c right) and *in silico* systems investigated in the previous section (Supplementary Fig. 5f). ...”

9. Fig. 4d-e dashed boxes: Why are regulations not inferred when they share a common target? Is it another limitation of the method?

We appreciate the reviewer for pointing out that 1D regulations are not inferred when they share a common target, which is an important aspect of the framework. Please note that this is not a limitation of GOBI, but rather a designed feature. Specifically, GOBI omits inferred regulations when they do not match the dimension of the framework. In the example of cofactors at the estrogen-sensitive *pS2* promoter (Fig. 4d), our 1D criteria $TRS_{X\sigma}^Y > TRS^{thres}$ inferred 1D regulations $HDAC \dashv hER$ and $TRIP1 \dashv hER$. These inferences indicate that *hER* is negatively regulated by two components *HDAC* and *hER*, which is a 2D regulation. However, since our 1D framework only guarantees a single cause for each target, we discarded these inferred 1D regulations. If both components are indeed effective together, i.e., 2D regulation $HDAC \dashv TRIP1 \dashv hER$ is present, those will be inferred from the 2D framework. However, with the 2D inference framework, the 2D regulation was not inferred from GOBI because $TRIP1 \dashv hER$ was identified as an indirect regulation. Thus, by merging results of 1D inference ($HDAC \dashv hER$ or $TRIP1 \dashv hER$ if *hER* is regulated by a single cause) and 2D inference (the absence of $TRIP1 \dashv hER$ regulation), we concluded that the inferred 1D regulation $HDAC \dashv hER$ is the only regulation. This is why we need to perform 2D inference rather than simply merging two 1D inferred regulations. This issue has been now more clearly described in the manuscript as follows:

- Section II. D: “GOBI infers five 1D regulations ($HDAC \dashv hER$, $TRIP1 \dashv hER$, $hER \rightarrow POLII$, $TRIP1 \dashv POLII$, and $HDAC \dashv POLII$) that satisfy the criteria $TRS_{X\sigma Y-}^Y > TRS^{thres}$. However, we exclude them because *hER* and *POLII* have two and three causes, forming 2D and 3D regulations, respectively, although the 1D criteria assumes a single cause (Fig. 4d middle, dashed box). If all of these regulations are effective, they will be identified as 2D and 3D

regulations. Indeed, among the 11 candidates for 2D regulations, most of them include the five inferred 1D regulations. Via Δ test and surrogate test, *indirect regulations are identified among inferred 2D regulations (Supplementary Fig. 9d). For example, $hER \rightarrow HDAC \dashv POLII$ satisfies the criteria $TRS_{X\sigma Y^-}^Y > TRS^{thres}$. Among two causal variables (i.e., hER and $HDAC$), only positive regulation from hER passes the post-filtering test, i.e., only 1D regulation $hER \rightarrow POLII$, but not $HDAC \dashv POLII$ is inferred as a direct regulation. Consequently, after excluding all the indirect regulations, two 1D regulations ($hER \rightarrow POLII$ and $HDAC \dashv hER$) and one 2D regulation $POLII \rightarrow TRIP1 \rightarrow HDAC$) are inferred.”*

- Section II. D: “... While two positive causal links from NO_2 and respirable suspended particulates ($Rspar$) to the disease are identified as 1D regulations (Fig. 4e middle), we exclude them because they share the same target (Fig. 4e middle, dashed box). Among two inferred 2D regulations, one passes the Δ test and surrogate test (Fig. 4d middle).”

Reviewer: 3

The authors propose a new method to detect regulations and their types in a class of systems suited to model biological oscillators. The methodology is applied successfully to both synthetic and experimental data. For this reason, I think it deserves to be published. However, I believe that a revision of the manuscript is necessary.

1. There is a clash between the authors' claim of a general inference method and the details provided in the text, which focuses mainly on computational biology and regulatory networks. For example, in the introduction, model-based inference is presented only focusing on algorithms designed for regulatory networks, and popular and powerful methods such as Kalman Filter (that requires only a model, not restricting it to any class, and can handle both measurement and dynamical noise) are not even mentioned. Interaction inference is a broad field. Methodology and literature are vast and stretch well beyond the papers highlighted by the authors, which are very specific to regulatory networks. The authors should either broaden their view throughout the paper, including applications, or tune it to the designed audience and preferred realm of application (starting from the title).

We thank the reviewer for highlighting the importance of including the Kalman Filter in the discussion of model-based inference. We have now discussed this in the Introduction section as follows:

- Section I: Alternatively, model-based methods infer causality by testing the reproducibility of time-series data with mechanistic models *using various methods such as simulated annealing (Gotoh, Tetsuya, et al., PNAS, 2016) and the Kalman Filter (Wang, Zidong, et al.; IEEE/ACM Transactions on Computational Biology and Bioinformatics, 2009, Pirgazi, J.; Khanteymooori, A. R. PloS one, 2018)*. Although testing the reproducibility is computationally expensive, as long as the underlying model is accurate, the model-based inference methods are accurate, even in the presence of synchrony in time series and indirect effect [21–29].

We agree with the reviewer's point that our paper focused on examples from biology, covering various scales. Specifically, we investigated gene regulatory networks such as the genetic oscillator (Fig. 4b), repressilator (Fig. 4c), and cofactors at the estradiol-sensitive promoter (Fig. 4d), as well as ecological systems such as the prey-predator system (Fig. 4a) and the impact of air pollutants on cardiovascular disease (Fig. 4e). Please note that previous publications on causation detection in *Nature Communications* have also focused on examples from biology (e.g., Leng, Siyang, et al., Nat Commun, 2020; Yang, A.C., Peng, CK. & Huang, N.E., Nat Commun, 2018; Orhan, A., Ma, W.J., Nat Commun, 2017). Given this context, we believe that our focus on biological examples aligns with the journal's aim and scope and does not mislead its readership.

2. The part of the Discussion section regarding the method limitations is very slim. There are limitations and assumptions here and there (such as thresholds when applying to noisy series) on which the authors should spend few words in the Discussion.

We agree with the reviewer that the limitations and assumptions of our method deserve more attention. Although we mentioned these limitations and assumptions throughout the manuscript, we recognize the need to explicitly address them in the Discussion section as the reviewer suggested. We have now clearly described the two limitations and two assumptions of our methods in the discussion section: the choice of threshold when applying to noisy data (please also look at response 7 for Reviewer 2), the choice of sampling rate (please also look at response 3 for Reviewer 1), the assumption about monotonic regulation (please also look at response 6 for Reviewer 1), and the assumption about negative self-regulation (please also look at the responses 4 and 5).

- Section III: “ ...

Despite these advantages, our method has some limitations that should be addressed. First, our framework assumes that when X causes Y , X causes Y either positively or negatively. Thus, GOBI cannot capture the regulation when X causes Y both positively and negatively or when the type of regulation changes over time. However, GOBI can be potentially extended to detect temporal-structured models, including non-monotonic regulation (Supplementary Fig. 11). It would be interesting in future work to investigate the extended framework thoroughly under diverse circumstances. Additionally, while we have considered the general form of monotonic ODE (Eq. (1)), GOBI can also be extended to describe interactions including time delays (Supplementary Fig. 12). This will be an interesting future direction to make GOBI more broadly applicable. Also, another limitation is the possibility of false-positive predictions since our method tests the reproducibility of time-series data using necessary conditions. To resolve this, we use multiple time-series data and perform post-filtering tests (i.e., Δ test and surrogate test). Nonetheless, it should be noted that inferring high-dimensional regulations requires a large amount of data (Supplementary Fig. 13). To address this challenge, we can use prior knowledge about the system. For example, in biological systems, negative self-regulation can be assumed, as the degradation rates of molecules increase as their concentrations increase. By assuming negative self-regulation, we are able to reduce the ND regulation to $(N - 1)D$ regulation, which allows us to successfully infer the network structure even with a small amount of experimental data (Fig. 4c). Note that when a priori assumption (e.g., the types of self-regulation) is not met, only the links that violate the assumptions are not trustable, i.e., the other inference results are not affected (Supplementary Fig. 3).

To use GOBI, we need to choose hyper-parameters. When applying GOBI to noisy data, users must choose thresholds for the regulation-detection region, regulation-detection function, and total-regulation score as well as two critical values of significance (i.e., p -values for Δ test and surrogate test). In this study, we determine these values by using noisy simulated data of various examples (Fig. 3 and Supplementary Fig. 5). Nevertheless, these values are effective when they are applied to experimental time-series data (Fig. 4 and 5). Thus, we have set those values of hyper-parameters as the default values of GOBI. However, the optimal threshold may

vary depending on the data characteristics, and users may need to adjust the thresholds based on the importance of avoiding false-positive or false-negative predictions. Another hyper-parameter that requires consideration is the choice of sampling rate. In this study, we used a sampling rate of 100 points per period after evaluating the trade-off between computational cost and accuracy. However, users can decrease or increase the sampling rate if the computation speed is too slow or if a higher level of accuracy is required, respectively.

3. The paper has several statements that are just too generous towards the method, such as “our approach completely resolves the fundamental limit of model-based inference: strong dependence on a chosen model.” The presented method can only be applied to a specific class of systems, so it sits somewhere between model-based inference and model-free inference. Furthermore, in the noisy case, it depends on two thresholds. Tuning down the text, eliminating these kinds of statements, and highlighting the limitation of the methodology does not take anything out of its value.

We completely agree with the reviewer’s concern that some of our statements were too generous in the current manuscript. In response, we have toned down the language to be more cautious in our claims as shown below. Also, we have expanded the Discussion section to include a more detailed explanation of the limitations of our approach (see Reviewer 3’s comment 2).

- Abstract: “... Here, we address this limitation by deriving an easily testable condition for a general *monotonic* ODE model to reproduce time-series data. We built a user-friendly computational package, GOBI (General ODE-Based Inference), which is applicable to nearly any *monotonic system with positive and negative regulations* described by ODE. ...”

- Section I: “Here, we develop a model-based method that infers interactions among multiple components described by the general *monotonic* ODE model:

$$\frac{dY}{dt} = f(X) = f(X_1, X_2, \dots, X_N),$$

where f can be any smooth and monotonic increasing or decreasing functions of X_i and X_N is Y in the presence of self-regulation. Thus, our approach *considerably* resolves the fundamental limit of model-based inference: strong dependence on a chosen model.”

- Section III: “We develop an inference method that *considerably resolves* the weakness of model-free and model-based inference methods. We derive the conditions for interactions satisfying the general *monotonic* ODE (Eq. (1)). As this allows us to easily check the reproducibility of given time-series data with the general *monotonic* ODE (i.e., the existence of ODE satisfying given time-series data) without fitting, the computational cost is dramatically reduced compared to the previous model-based approaches. Importantly, as our method can be applied to any system described by general *monotonic* ODE (Eq. (1)), it *significantly addresses*

the fundamental limit of the model-based approach (i.e., requirement of a priori model accurately describing the system) (*Supplementary Fig. 4*). ...”

4. “In most biological systems, the degradation rates of molecules increase as their own concentrations increase; thus we assume that self-regulation is negative for every component in the system”. This is another assumption that was not mentioned in presenting the method. If the method is good, this assumption is not needed. On the contrary, it would be an output of the inference. The authors should show that.

GOBI can infer regulations without the assumption of negative self-regulation as it can also detect whether the self-regulation is positive or negative. Thus, the assumption for negative self-regulation is optional and can be used to reduce the computational cost and data requirements. Similarly, if we have any prior knowledge of the part of regulatory networks, we can assume specific regulation types for the subnetworks based on the prior knowledge. To aid the reader's understanding, we have now discussed this in the Supplementary Information as follows:

- Supplementary Information:

Section VI. Incorporating prior knowledge into the inference from GOBI

*When using GOBI for inference, users have the option to incorporate prior knowledge into the analysis. This involves assuming the types of regulation for specific parts of the network based on any available prior knowledge they have, which allows effective inference with a limited amount of data. In particular, in biological systems, the degradation rates of molecules typically increase as their own concentrations increase, so users can often assume negative self-regulation, as described in our manuscript. Of course, wrong assumptions may result in some incorrect inference results. To illustrate this, we use the example of the Frzilator (*Fig. 2c*) to infer the network structure with and without assuming the types of self-regulation.*

*Using the Frzilator ODE model, we first simulate one, five, and ten time series from different initial conditions which lie in the range of the original limit cycle (*Supplementary Fig. 3a (i)-(iii)*). From the time-series data, GOBI successfully infers the true network structure regardless of the amount of data when we assume the negative self-regulation (*Supplementary Fig. 3b, the first row*). However, without assuming self-regulation (*Supplementary Fig. 3b, the second row*), the inference results differ depending on the amount of data. Specifically, when ten time-series data are used, GOBI successfully infers the network structure. However, as the amount of data used decreases, the inferred network has some incorrect predictions (*Supplementary Fig. 3b, red arrows in the second row*). Taken together, assuming the types of self-regulation is not necessary, but it can be a useful option when the amount of available data is limited.*

*However, making wrong assumptions about the types of self-regulation can lead to some incorrect inference results. To investigate this, we apply GOBI by falsely assuming a positive self-regulation of f and correctly assuming negative self-regulations of c and e (*Supplementary Fig. 3b, the third row*). While GOBI successfully infers $f \rightarrow c$ and $c \rightarrow e$, GOBI fails to infer*

regulation $e \dashv f$. This indicates that GOBI can incorrectly infer the regulation targeting the component whose self-regulation is incorrectly assumed.

Supplementary Fig. 3. Inference of network structures under different assumptions of self-regulation types. a The Frzillator ODE model (Fig. 2b) is used to simulate one (a (i)), five (a (ii)), and ten (a (iii)) time series from different initial conditions. **b** Using these time-series data, the inference results from GOBI with three different approaches are illustrated: assuming negative self-regulations (b the first row), not assuming the types of self-regulation (b the second row), and assuming the wrong types of self-regulation (b the third row), where dashed lines indicate the assumed self-regulation types. In the last approach, the positive self-regulation of f is assumed, which is incorrect. Assuming negative self-regulation allows GOBI to accurately infer the network structure regardless of the amount of data (b the first row). However, not assuming the types of self-regulation leads to false-positive predictions (red arrows) when the amount of data is limited (b the second row). Under wrong assumptions of self-regulation types, GOBI only infers regulations targeting components whose self-regulations are correctly assumed (b the third row).

Also, we have now revised our manuscript as follows:

Section II. B: "... We apply the framework to infer regulatory networks from simulated time-series data of various biological models. In these models, the degradation rates of molecules increase as their concentrations increase, like in most biological systems (i.e., self-regulation is negative). Such prior information, including the types of self-regulation, can be incorporated into our framework. For example, to incorporate negative self-regulation, when detecting N D regulation, one can use the $(N + 1)D$ regulation-detection function and score that include negative self-regulation. Specifically, when inferring $1D$ positive regulation from X to Y , the criteria $S_{X+Y}^Y = 1$ is used. To illustrate this, we assume the negative self-regulation to infer the network structures of biological models (see below for details). Note that this assumption is optional for inference (see Supplementary Information for details).

...

Here, assuming negative self-regulation allows us to reduce N D regulation to $(N-1)D$ regulation (Supplementary Fig. 3). This simplification is important for accurate inference when data is limited. Moreover, it should be noted that when the assumptions about the types of self-

regulation are not met, only the links that violate these assumptions become untrustworthy, while the other inference results are not affected (Supplementary Fig. 3). ...”

Previously, when applying GOBI to the synthetic genetic oscillator example, we assumed negative self-regulation (Fig. 4b). However, as experimental data is sufficient for this case, we no longer need to make this assumption. Thus, we now apply GOBI without assuming negative self-regulation as follows:

Section II. D: “... Next, we apply GOBI to the time series of the synthetic genetic oscillator, which consists of Tetracycline repressor (*TetR*) and RNA polymerase sigma factor (σ^{28}) [43] (Fig. 4b left). While the time series are measured under different conditions after adding purified *TetR* or inactivating intrinsic *TetR*, our method consistently infers the negative feedback loop including negative self-regulations based on two direct regulations $\sigma^{28} \rightarrow TetR$ and $TetR \dashv \sigma^{28}$ for all cases (Fig. 4b middle and Supplementary Fig. 9b). This indicates that our method can infer regulations even when the data are achieved from different conditions since we do not specify the specific equations with parameters in Eq. (1).

We next investigate the time-series data from a slightly more complex synthetic oscillator, the three-gene repressilator [44] (Fig. 4c left). As the amount of data is greatly reduced compared to the synthetic genetic oscillator (Fig. 4b), we assume self-negative regulation.

...

Furthermore, compared to the synthetic genetic oscillator (Fig. 4b), the amount of data is small and the number of components is large; thus, it is essential to assume negative self-regulation for correct inference, i.e., without the assumption, the available data is insufficient to fill the space of the regulation-detection function, making it difficult to detect 2D regulations.”

• Supplementary Information:

Section XII: “... Similarly, for the genetic oscillator, the criteria of TRS for 1D regulation infers two direct regulations, $TetR \dashv \sigma^{28}$ and $\sigma^{28} \rightarrow TetR$ (Supplementary Fig.9b (i)). Then, we check the 2D regulations including self-regulation. Two 2D regulations, $TetR \dashv \sigma^{28} \rightarrow TetR$ and $TetR \dashv \sigma^{28} \dashv \sigma^{28}$, are inferred using the criteria $TRS_{X\sigma_Y}^Y > TRS^{thres}$ (Supplementary Fig. 9b (ii)). Both regulations pass the Δ test, and we do not perform the surrogate test as there are no possible indirect regulations in the two-component system. This indicates that assumptions incorporating prior information (such as the types of self-regulation) are unnecessary when sufficient data is present. ...”

b. Genetic oscillator

5. What happens in systems where the assumptions are not met? Is all inference messed up? Or only the links violating the assumptions are not trustable?

We thank the reviewer for this comment. When the assumptions about the types of self-regulations are not met, only the links violating the assumptions are not trustable. We have now discussed this in the Supplementary Information (Supplementary Fig. 3 and see Reviewer 3's comment 4 for details).

6. If I well understand, the thresholds for the network inference are derived a posteriori to optimize the inference performance. If so, this should be discussed as a limitation, at least in the Discussion. Which is the sensitivity of the results to the threshold values? And how should the user proceed to choose the thresholds if ground-truth data is not available?

We do not need the ground-truth to choose the thresholds. Specifically, to run inference on noisy time-series data using GOBI, users need to choose three different thresholds for regulation-detection region, regulation-detection score, and total-regulation score. Thus, we suggested guidelines for selecting appropriate thresholds based on the noise level and the dimension of the regulation using simulated time-series data obtained from various ODE systems (Fig. 3a-c and Supplementary Fig. 5). Specifically, our guidelines involve decreasing the thresholds for regulation-detection score and total regulation score as the noise level increases. Note that the noise level can be quantifiable with sole time series (see Methods and Supplementary Fig. 8). Additionally, our guidelines involve decreasing the threshold for regulation-detection region as the dimension of regulation increases (see Supplementary Information for details). As we used these default guidelines of GOBI for the inference from experimental data (Fig. 4), we did not use the experimental data to derive posteriori thresholds for optimizing the inference performance. Specifically, we estimate the noise level in experimental data sets, determine the dimension of regulation that we want to infer, and then apply our guidelines to choose appropriate thresholds accordingly. Indeed, the guidelines derived from simulated data were effective for the experimental data sets as well (Fig. 4 and Supplementary Fig. 9). Also, to test the robustness of our guidelines, we have illustrated the sensitivity of the results to changes in threshold values (Supplementary Fig. 5). Thus, by using the guidelines we have suggested, users can select appropriate threshold values even in the

absence of ground-truth data. Of course, users can also adjust the thresholds depending on whether the goal is decreasing false-positive or false-negative predictions. Nevertheless, we agree with the reviewer's comment regarding the limitations of our proposed thresholds and have explicitly addressed them in the Discussion section (see Reviewer 3's comment 2). Furthermore, we have provided an explicit explanation for our choice of thresholds in the Results section:

• Section II D (page 6): *“When the proposed thresholds for the regulation-detection score (Fig. 3b) and Total Regulation Score (Fig. 3c) and two critical values of significance (i.e., p -value = 0.01 for the Δ test and p -value = 0.001 for the surrogate test) are used, GOBI successfully infers the regulatory networks from in silico time series. Here, we use GOBI with these default hyperparameters to infer regulatory networks from experimentally measured time series. ...”*

Some additional minor points:

7. Sec II A: “positive or negative causation” maybe the authors meant “regulation” rather than “causation”

In this study, we defined causation as direct regulation, which means that if X causes Y , then the dynamic of Y is given as $\dot{Y} = f(X)$, where f is a monotonic function with respect to X . Consequently, in our manuscript, we used both ‘causation’ and ‘regulation’ to refer to this concept, which could be confusing for readers. To address this issue, we have now replaced all instances of ‘causation’ with ‘regulation’ in the Results section to clarify our terminology.

8. Sec II A: The authors should state clearly how Δ is extended to three and more dimensions. The reader can get the idea but being explicit is always better.

We agree with the reviewer's comment regarding the need for a clear explanation of how the Δ test is extended to three or more dimensions. In our current manuscript, we have only provided a general definition of the Δ test for higher dimensions without further elaboration. To improve the reader's understanding of the Δ test for high dimensions, we have now provided explicit explanations including a simple example as follows:

• Supplementary Information

Section I. B (page 4): *“... For example, let us consider the 3D regulation $X_1 \rightarrow X_2 \dashv X_3 \rightarrow Y$, which has been inferred from the criteria of regulation-detection score, $S_{X_1^+ X_2^- X_3^+}^Y = 1$. To determine whether it is a false-positive prediction or not, we need to perform three Δ tests, each focusing on a different causal variable. Specifically, we compute $\Delta_{X_1^+ X_2^-}^Y(X_3) := S_{X_1^+ X_2^- X_3^+}^Y - S_{X_1^+ X_2^- X_3^-}^Y$, $\Delta_{X_1^+ X_3^+}^Y(X_2) := S_{X_1^+ X_2^+ X_3^+}^Y - S_{X_1^+ X_2^- X_3^+}^Y$, and $\Delta_{X_2^- X_3^+}^Y(X_1) := S_{X_1^+ X_2^- X_3^+}^Y - S_{X_1^- X_2^- X_3^+}^Y$. If they are not zero for some data, then the regulation $X_1 \rightarrow X_2 \dashv X_3 \rightarrow Y$ pass the Δ tests. However, if $\Delta_{X_1^+ X_2^-}^Y(X_3)$ is zero for all data, $X_3 \rightarrow Y$ is a false-positive because X_3 does not affect an existing*

regulation $X_1 \rightarrow X_2 \dashv Y$. Similarly, if $\Delta_{X_1^+ X_3^+}^Y(X_2)$ and $\Delta_{X_2^- X_3^+}^Y(X_1)$ are zero, then $X_2 \dashv Y$ and $X_1 \rightarrow Y$ are false-positive predictions, respectively.”

9. Sec IIC: The noise in the time series is measurement noise, not dynamical, isn't it? It should be stated clearly in the text. Also, it would be interesting to see, at least for a simple negative feedback loop of two nodes, what would be the impact of dynamical noise.

As the reviewer mentioned, in our current manuscript, the noise in the time series is measurement noise and this needs to be explicitly stated in the manuscript. To address this concern, we have revised the manuscript as follows:

Section IV. B: “... *To introduce measurement noise in time series*, we introduce multiplicative noise sampled randomly from a normal distribution with mean 0 and standard deviation given by the noise level. For example, for 10% multiplicative noise, we add the noise $X(t_i) \cdot \varepsilon$ to $X(t_i)$, where $\varepsilon \sim N(0, 0.1^2)$”

Also, to make GOBI more widely applicable, it is crucial to consider the influence of dynamical noise. Thus, we have now illustrated the impact of dynamical noise on the performance of GOBI (Supplementary Fig. 7). Our findings indicate that GOBI exhibits a comparable level of robustness to dynamical noise as it does to measurement noise (see Reviewer 1's comment 2).

10. How much data is needed to run the inference? How does it depend on the order of the interaction? Is it suitable for large networks? I understand that the space to compute S goes down exponentially but I think the paper needs more quantitative statements

We appreciate the reviewer's comment. As the amount of data increases, the confidence in the inference also increases. Furthermore, not only the quantity but also the quality of the data is important. In particular, we found that using multiple short time-series data from different initial conditions can lead to more accurate results compared to using a single long-time series (Fig. 4b). As both the quantity and quality of the time-series data are essential for accurate inference, it is difficult to provide a simple rule for determining how much data is needed to run the inference. However, one guideline we can suggest to the user is to check how much the domain of the regulation-detection function is filled with the data. As this space becomes more filled, GOBI can provide more confident results. We have now revised the computational package, GOBI, to show how much the domain of the regulation detection is filled (see Reviewer 2's comment 4 and Supplementary Fig. 14 for details). In particular, if this filling is not enough to detect to a specific dimension of regulation, GOBI provides a warning signal to users. Furthermore, we have now illustrated this guideline using simple examples in the Supplementary Information as follows:

- Supplementary Information

Section XVI. Guideline for determining whether the data is sufficient or not

Here, we suggest a guideline to assist users in determining the amount of data required for GOBI with simple examples. With the ODE describing the regulation (Supplementary Fig. 13a-c):

$$\dot{Y} = \sum_{i=1}^d X_i,$$

where d is the dimension of regulation, we simulate 50 time series on $[0, 1]$ from different initial conditions. The time series of input signal X_i are constructed by connecting 5 randomly selected points from $[0, 1]$ over the time domain with the spline fitting. Among the 50 time-series data, we use part of them (i.e., M number of data, Supplementary Fig. 13d-f (i)-(v)) to compute the regulation-detection function ($I_{X^+}^Y$). Then, we focus on its domain, the regulation-detection region (R_{X^+}). As we increase the number of time-series data, the space of the regulation-detection function gradually expands. Once the space fills the entire domain, GOBI can provide a confident result. Achieving full coverage of the domain requires exponentially more data as the dimension of regulation increases, assuming a similar quality of data. For example, when using 10, 25, and 50 numbers of data, the space of $I_{X^+}^Y$ for 1D, 2D, and 3D regulations are similarly filled, respectively (Supplementary Fig. 13b (iii), c (iv), and d (v)). Thus, in this example, twice as much data is required as the dimension increases by one. In case the available data is not sufficient, i.e., when the domain is not adequately filled, our framework will issue a warning that more data is required for confident result.

Supplementary Fig. 13. The required amount of data increases as the dimension of regulation increases. **a-c** Three different dimensional regulations from X to Y : 1D regulation (**a**), 2D regulation (**b**), and 3D regulation (**c**). For each regulation, 50 time-series data are simulated on $[0, 1]$ from different initial conditions. **d-f** Using the part of the time-series data (M number of

data, $\mathbf{d-f}$ (i)-(v)), $I_{X^+}^Y$ are computed and their domains (i.e., regulation-detection region, R_{X^+}) are shown. As the number of data increases, the space of the regulation-detection function gradually expands and fills the entire domain ($[0, 1]^2$) if there is enough data. To fill the entire domain, exponentially more data is required as the dimension increases (\mathbf{d} (iii), \mathbf{e} (iv), and \mathbf{f} (v)).

11. Since the authors present an implementation, it would be nice to know the order of magnitude of the runtime

Now, we have discussed the order of magnitude of the runtime in the Supplementary Information as follows:

- Supplementary Information
Section XVII. Computational cost of GOBI

In this section, we provide the order of magnitude of runtime for GOBI, assuming N components in the system, M time-series data, and T time points for each time series. To infer every d -dimensional regulation in the system, we perform three steps in GOBI: regulation-detection score, delta test, and surrogate test. First, from a single time-series data, the computation of the regulation-detection score has a time complexity of $O(T^2)$. When considering all possible d -dimensional regulations and using M time-series data, the computation of regulation-detection scores has a time complexity of $O(M \times N^d \times T^2)$. Next, in the delta test, for each regulation that satisfies the criteria of the regulation-detection score, we perform the Wilcoxon signed-rank test on every causal variable. Since the Wilcoxon signed-rank test has a time complexity of $O(M^2)$, the delta test has a time complexity of $O(M^2 \times d)$ for a single regulation. Thus, the worst-case time complexity of the delta test is $O(M^2 \times d \times N^d)$. Lastly, in the surrogate test, for each regulation that satisfies the criteria of the delta test, we compute the regulation-detection score with surrogate time series. If the number of surrogate time series is S , then the surrogate test has a time complexity of $O(T^2 \times S \times M)$ for a single regulation. Thus, the worst-case time complexity of the surrogate test is $O(T^2 \times S \times M \times N^d)$. Note that our method can reduce the runtime by utilizing parallel computing.

12. In the supplementary material, the description of the noise level calculation could be improved. Just saying that the library uses a MATLAB function (of which I couldn't find the documentation) without mentioning what the function does is a bit short, being that an ingredient of the calculation. Same for the MATLAB function "gradient". Is it applied before or after the noise filtering? I guess after, but it should be specified. And what is the underlying algorithm?

We agree with the reviewer that our current manuscript does not fully describe the use of the MATLAB functions. In response to the reviewer's concern, we have now revised the manuscript as follows:

- Section IV. A: "Here, we describe the key steps of our computational package, GOBI (Github link will be provided upon acceptance). For the experimental time-series data $X(t) =$

$(X_1(t), X_2(t), \dots, X_N(t)), X(t)$ can be interpolated with either the 'spline' or 'fourier' method, chosen by the user. *For the spline interpolation, we use MATLAB function 'interp1' with an option 'spline', and for the fourier interpolation, we use MATLAB function 'fit' with an option 'fourier1-8'. After the interpolation, the derivative of $X(t)$ is computed using the MATLAB function 'gradient' to compute the regulation-detection score.*"

- Supplementary Information (Section XII):

"Supplementary Fig. 8. Approximate noise level of experimental data using residual. **a** For each in silico system, we compute the mean square of the residual between noisy and fitted time series when the noise level varies. Here, fitted time series are obtained by using the MATLAB function 'fit' with an option 'fourier4'. The mean square of the residual is averaged over all components in the system and we simply called it 'residual'. The value of the residual increases as the noise level increases and their tendency is similar among the systems. **b** Using this tendency, we can approximate the noise level of experimental data. For each system that we used (prey-predator, genetic oscillator, repressilator, estradiol data set and air pollutants and cardiovascular disease), *experimental time-series data are interpolated using the MATLAB function 'fit' with an option 'fourier4'. Then, we compute the residuals and approximate their noise level.*"

13. In the calculation of GC, I couldn't find how you chose the order of the AR processes for Y and X.

In the calculation of GC, we unintentionally omitted the order of the AR processes. We have now revised the Methods section to include the order of the AR processes (see Reviewer 3's comment 14).

14. Speaking of GC, if you shift one series and test GC $X(t) \square X(t+T)$ (Fig.4ab) and GC does not detect that as a link, I would be surprised, as the shifted timeseries becomes an AR process of its past, which is exactly what GC tests for. Maybe the authors could discuss more in depth why they chose to employ this test. Alternatively, I think that if they want a null hypothesis for non-connection, it could have more sense to use a series from another dataset (such as TetR & P & D).

We thank the reviewer for this comment. We performed the GC using the code provided in [Chandler, 2020.], and specified the maximum order of AR process that produced the first minimum of delayed mutual information. For oscillatory time series, the delayed mutual information typically exhibits its first minimum at a quarter of the period. In our examples (Fig. 5a-b), each time series of the prey-predator system and genetic oscillator was duplicated and shifted about half of its period. This indicates that the shifted time series does not become an AR process of its past. Thus, it is possible that GC does not detect the link from the original time series to the shifted one.

Nevertheless, we agree with the reviewer's point that the examples of the shifted system were not suitable for testing the model-free methods. When the order of AR process exceeds half of the period, the shifted time series becomes an AR process of its past, and the GC automatically identifies the link from the original time series to the shifted one. Also, shifting the time series does not significantly alter its shadow manifold; thus, PCM and CCM are likely to identify the link between the original and shifted time series. Taking into account these concerns, we have revised our approach by using a new example to compare the performance of GOBI and model-free methods.

As recommended by the reviewer, we have merged the time series of two different data sets instead of shifting them. Specifically, from the set of eight different time-series data of genetic oscillator measured under different conditions, we selected one that has a similar phase to the time series of the prey-predator system. Then, the selected time series was merged with the time series of the prey-predator system. Using this merged system, we tested GOBI and model-free methods. We found that both PCM and GOBI successfully inferred the true network structure, whereas CCM and GC did not. Furthermore, when we reduced the sampling rate by half, the accuracy of PCM dramatically dropped, whereas GOBI was still able to infer the true network structure (see Reviewer 1's comment 3). This finding indicates that model-free methods often misidentify synchrony for causality.

Based on this, we have revised the manuscript as follows:

Section II. E: *"For the prey-predator system and the genetic oscillator (Fig. 4a-b), we merge them to create a more challenging case (Fig. 5a). Specifically, from the set of eight different time-series data of a genetic oscillator measured under different conditions, we select one that has a similar phase to the time series of the prey-predator system (Fig. 4b panel at the 2nd row and 2nd column). Then, we merge the selected time series with the time series of the prey-predator system. While GOBI and PCM successfully detect two independent feedback loops (Fig. 5a), CCM and GC infer false positive predictions (e.g., P to σ^{28} in Fig. 5a) because they usually misidentify synchrony as causality. Furthermore, when we reduce the sampling rate by half, the accuracy of PCM dramatically drops, whereas GOBI can still infer the true network structure (Supplementary Fig. 10)."*

Fig. 5. Model-free methods, but not our method, make a false prediction due to the presence of synchrony and indirect effect. **a-b** We apply our method and popular model-free methods (i.e., GC, CCM, and PCM) to various experimental time-series data obtained from the *prey-predator system merged with the genetic oscillator* (**a**); *repressilator* (**b**); *cofactors at the pS2 promoter* (**c**); and *air pollutants and cardiovascular disease* (**d**). For the air pollutants and cardiovascular disease data, we test the methods on three years of data (**d** grey) and on two years of data (**d** purple).

• Section IV. A5: “For Convergent Cross Mapping (CCM) [3] and Partial Cross Mapping (PCM) [20], we choose an appropriate embedding dimension using the false nearest neighbor algorithm. Also, we select a time lag producing the first minimum of delayed mutual information. To select the threshold value T in PCM, we use k -means clustering as suggested in [20]. For Granger Causality (GC) [2], we run the code provided in (Chandler, 2020), specifying the order of AR processes of the first minimum of delayed mutual information as we choose a max delay with the CCM and PCM. Also, we reject the null hypothesis that Y does not Granger cause X , and thereby infer direct regulations by using the F statistic with a significance level of 95% (Granger, *Journal of Econometric Society*, 1969).”

Reviewer #1 (Remarks to the Author):

This reviewer thanks the effort made by the authors, thoroughly addressing my concerns as well as the other experts' concerns in the first round of review. Currently, my only concerns are listed below.

1. Clearly, the proposed method does not rely on the specific form of the model. As such, this method could be seen as a model-free algorithm NOT a model-based technique yet. Actually, all the data-driven methods use an assumption that there is a hidden dynamics generating the data that are collected by the experiments.

2. In the title, the authors emphasize "the curse of synchrony...". In my opinion, the current method still cannot deal with the case of complete synchronization dynamics. I therefore suggest to correct as "the curse of generalized-synchrony...". However, this is not compulsory. Alternatively, the authors could mention this point in the discussion or in the concluding remarks.

Reviewer #2 (Remarks to the Author):

The authors have revised their manuscript with additional analyses and substantial consideration, which improved the quality and readability of their work. The authors have responded to my original comments in detail, and have addressed the most important questions I was raising (i.e. the monotonicity assumption).

Therefore, I recommend publication.

Reviewer #3 (Remarks to the Author):

I believe the authors have addressed the issues I raised in my previous review.

Two minor things I spotted in the updated manuscript:

- a) Pag. 9 "circumferences": Perhaps the authors meant "circumstances"?
- b) Pag. 9: "Also, another limitation is the possibility of false-positive predictions since our method tests the reproducibility of time-series data using necessary conditions." I think this sentence is rather obscure and should be rephrased.

Reviewer: 1

This reviewer thanks the effort made by the authors, thoroughly addressing my concerns as well as the other experts' concerns in the first round of review. Currently, my only concerns are listed below.

1. Clearly, the proposed method does not rely on the specific form of the model. As such, this method could be seen as a model-free algorithm NOT a model-based technique yet. Actually, all the data-driven methods use an assumption that there is a hidden dynamics generating the data that are collected by the experiments.

As the reviewer mentioned, in this paper, all inferences were performed without making assumptions about the explicit form of the underlying ODE. However, our approach assumes that all the interactions can be described by a monotonic ODE. Based on this assumption, we classify our approach as a general model-based method.

2. In the title, the authors emphasize "the curse of synchrony...". In my opinion, the current method still cannot deal with the case of complete synchronization dynamics. I therefore suggest to correct as "the curse of generalized-synchrony...". However, this is not compulsory. Alternatively, the authors could mention this point in the discussion or in the concluding remarks.

We agree with the reviewer's point that GOBI still cannot address the case of complete synchronization dynamics. To help the reader's understanding, we have now revised the introduction and discussion section as follows:

- Section I: "... Various model-free methods, such as Granger Causality (GC) [2] and Convergent Cross Mapping (CCM) [3], have been widely used to infer causation from time-series data. Although they are easy to implement and broadly applicable [4–10], they usually struggle to differentiate *generalized*-synchrony (i.e., similar periods among components) versus causality [11–15] and distinguish between direct and indirect causation [16–20]. ..."
- Section I: "... Although testing the reproducibility is computationally expensive, as long as the underlying model is accurate, the model-based inference method is accurate even in the presence of *generalized*-synchrony in time series and indirect effect [21–29]. ..."
- Section III: "... Importantly, as our method can be applied to any system described by general monotonic ODE (Eq. (1)), it significantly addresses the fundamental limit of the model-based approach (i.e., requirement of a priori model accurately describing the system) (Supplementary Fig. 4). In addition, our method also does not run the serious risk of misidentifying *generalized*-synchrony as causality, unlike the previous model-free approaches. *Please note that our approach still cannot deal with complete synchronized system.* ..."

Reviewer: 2

The authors have revised their manuscript with additional analyses and substantial consideration, which improved the quality and readability of their work. The authors have responded to my original comments in detail, and have addressed the most important questions I was raising (i.e. the monotonicity assumption). Therefore, I recommend publication.

We appreciate the reviewer for valuable comments that have greatly improved our work.

Reviewer: 3

I believe the authors have addressed the issues I raised in my previous review.

Two minor things I spotted in the updated manuscript:

a) Pag. 9 “circumferences”: Perhaps the authors meant “circumstances”?

We thank the reviewer for bringing to our attention the misuse of the word. We meant “circumstances”, and have corrected the mistake in the manuscript.

b) Pag. 9: "Also, another limitation is the possibility of false-positive predictions since our method tests the reproducibility of time-series data using necessary conditions." I think this sentence is rather obscure and should be rephrased.

We agree with the reviewer that the sentence may confuse the reader. Based on this concern, we have now rephrased the sentence as follows:

- Section III: “... Also, another limitation is the possibility of false-positive predictions. *This occurs because* our method tests the reproducibility of time-series data using necessary conditions. *Specifically, the regulation-detection score can be one even in the absence of regulation. ...*”